# ContraNorm: A Contrastive Learning Perspective on Oversmoothing and Beyond

**Xiaojun Guo**[1*]  **Yifei Wang**[2*]  **Tianqi Du**[2*]  **Yisen Wang**[1,3 †]
[1]National Key Lab of General Artificial Intelligence,
  School of Intelligence Science and Technology, Peking University
[2]School of Mathematical Sciences, Peking University
[3]Institute for Artificial Intelligence, Peking University

## ABSTRACT

Oversmoothing is a common phenomenon in a wide range of Graph Neural Networks (GNNs) and Transformers, where performance worsens as the number of layers increases. Instead of characterizing oversmoothing from the view of *complete collapse* in which representations converge to a single point, we dive into a more general perspective of *dimensional collapse* in which representations lie in a narrow cone. Accordingly, inspired by the effectiveness of contrastive learning in preventing dimensional collapse, we propose a novel normalization layer called **ContraNorm**. Intuitively, ContraNorm implicitly shatters representations in the embedding space, leading to a more uniform distribution and a slighter dimensional collapse. On the theoretical analysis, we prove that ContraNorm can alleviate both complete collapse and dimensional collapse under certain conditions. Our proposed normalization layer can be easily integrated into GNNs and Transformers with negligible parameter overhead. Experiments on various real-world datasets demonstrate the effectiveness of our proposed ContraNorm. Our implementation is available at `https://github.com/PKU-ML/ContraNorm`.

## 1 INTRODUCTION

Recently, the rise of Graph Neural Networks (GNNs) has enabled important breakthroughs in various fields of graph learning (Ying et al., 2018; Senior et al., 2020). Along the other avenue, although getting rid of bespoke convolution operators, Transformers (Vaswani et al., 2017) also achieve phenomenal success in multiple natural language processing (NLP) tasks (Lan et al., 2020; Liu et al., 2019; Rajpurkar et al., 2018) and have been transferred successfully to computer vision (CV) field (Dosovitskiy et al., 2021; Liu et al., 2021; Strudel et al., 2021). Despite their different model architectures, GNNs and Transformers are both hindered by the oversmoothing problem (Li et al., 2018; Tang et al., 2021), where deeply stacking layers give rise to indistinguishable representations and significant performance deterioration.

In order to get rid of oversmoothing, we need to dive into the modules inside and understand how oversmoothing happens on the first hand. However, we notice that existing oversmoothing analysis fails to fully characterize the behavior of learned features. A canonical metric for oversmoothing is the average similarity (Zhou et al., 2021; Gong et al., 2021; Wang et al., 2022). The tendency of similarity converging to 1 indicates representations shrink to a single point (complete collapse). However, this metric can not depict a more general collapse case, where representations span a low-dimensional manifold in the embedding space and also sacrifice expressive power, which is called dimensional collapse (left figure in Figure 1). In such cases, the similarity metric fails to quantify the collapse level. Therefore, we need to go beyond existing measures and take this so-called *dimensional collapse* into consideration. Actually, this dimensional collapse behavior is widely discussed in the contrastive learning literature (Jing et al., 2022; Hua et al., 2021; Chen & He, 2021; Grill et al., 2020), which may hopefully help us characterize the oversmoothing problem of GNNs and Transformers. The main idea of contrastive learning is maximizing agreement between different augmented views of the same data example (i.e. positive pairs) via a contrastive loss. Common contrastive loss can be decoupled into alignment loss and uniformity loss (Wang & Isola, 2020).

---

*Equal Contribution.
†Corresponding Author: Yisen Wang (yisen.wang@pku.edu.cn)

The two ingredients correspond to different objectives: alignment loss expects the distance between positive pairs to be closer, while uniformity loss measures how well the embeddings are uniformly distributed. Pure training with only alignment loss may lead to a trivial solution where all representations shrink to one single point. Fortunately, the existence of uniformity loss naturally helps solve this problem by drawing representations evenly distributed in the embedding space.

Given the similarities between the oversmoothing problem and the representation collapse issue, we establish a connection between them. Instead of directly adding uniformity loss into model training, we design a normalization layer that can be easily used out of the box with almost no parameter overhead. To achieve this, we first transfer the uniformity loss used for training to a loss defined over graph node representations, thus it can optimize representations themselves rather than model parameters. Intuitively, the loss meets the need of drawing a uniform node distribution. Following the recent research in combining optimization scheme and model architecture (Yang et al., 2021; Zhu et al., 2021; Xie et al., 2021; Chen et al., 2022), we use the transferred uniformity loss as an energy function underlying our proposed normalization layer, such that descent steps along it corresponds with the forward pass. By analyzing the unfolding iterations of the principled uniformity loss, we design a new normalization layer **ContraNorm**.

As a proof of concept, Figure 1 demonstrates that ContraNorm makes the features away from each other, which eases the dimensional collapse. Theoretically, we prove that ContraNorm increases both average variance and effective rank of representations, thus solving complete collapse and dimensional collapse effectively. We also conduct a comprehensive evaluation of ContraNorm on various tasks. Specifically, ContraNorm boosts the average performance of BERT (Devlin et al., 2018) from 82.59% to 83.54% on the validation set of General Language Understanding Evaluation (GLUE) datasets (Wang et al., 2019), and raises the test accuracy of DeiT (Touvron et al., 2021) with 24 blocks from 77.69% to 78.67% on ImageNet1K (Russakovsky et al., 2015) dataset. For GNNs, experiments are conducted on fully supervised graph node classification tasks, and our proposed model outperforms vanilla Graph Convolution Network (GCN) (Kipf & Welling, 2017) on all depth settings. Our contributions are summarized as:

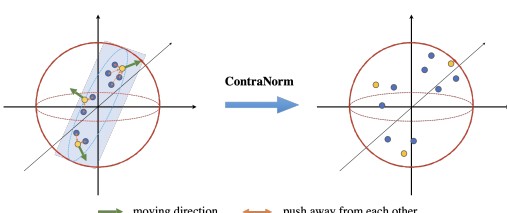

Figure 1: An illustration of how our proposed ContraNorm solves the dimensional collapse. **Left**: Features suffer from dimensional collapse. **Right**: With the help of ContraNorm, features become more uniform in the space, and the dimensional collapse is eased.

- We dissect the limitations of existing oversmoothing analysis, and highlight the importance of incorporating the dimensional collapse issue into consideration.

- Inspired by the techniques from contrastive learning to measure and resolve oversmoothing, we propose ContraNorm as an optimization-induced normalization layer to prevent dimensional collapse.

- Experiments on a wide range of tasks show that ContraNorm can effectively mitigate dimensional collapse in various model variants, and demonstrate clear benefits across three different scenarios: ViT for image classification, BERT for natural language understanding, and GNNs for node classifications.

## 2   BACKGROUND & RELATED WORK

**Message Passing in Graph Neural Networks.** In the literature of GNNs, message-passing graph neural networks (MP-GNNs) are intensively studied. It progressively updates representations by exchanging information with neighbors. The update of node $i$'s representation in $l$-th layer is formalized as $\boldsymbol{h}_i^{(l)} = \text{UPDATE}\big(\boldsymbol{h}^{(l-1)}, \text{AGGREGATE}(\boldsymbol{h}_i^{(l-1)}, \{\boldsymbol{h}_j^{(l-1)} \mid j \in \mathcal{N}(i)\})\big)$, where $\mathcal{N}(i)$ denotes the neighborhood set of node $i$, AGGREGATE$(\cdot)$ is the procedure where nodes exchange message, and UPDATE$(\cdot)$ is often a multi-layer perceptron (MLP). A classical MP-GNNs model is GCN (Kipf & Welling, 2017), which propagates messages between 1-hop neighbors using an adjacency matrix.

**Self-Attention in Transformers.** Transformers encode information in a global scheme with self-attention as the key ingredient (Vaswani et al., 2017). Self-attention module re-weights interme-

diate representations by aggregating semantically near neighbors. Formally, it estimates similarities between key and query, namely self-attention matrix, as $\bar{A} = \text{softmax}(QK^\top)$, $Q = XW_Q$, and $K = XW_K$ where $X$, $W_Q$, and $W_K$ are the input, weight matrices for query and key, respectively. A multi-head self-attention module with a residual connection can be formularized as $\text{attn}(X) = X + \sum_{k=1}^h \bar{A}_k XV_k W_k^\top$, where $h$ is the number of heads, and $V$, $W$ are weights for value and final output.

**Connections between Message Passing and Self-Attention.** Note that the self-attention matrix can be seen as a normalized adjacent matrix of a corresponding graph (Shi et al., 2022). Considering a weighted fully connected graph $G$ with adjacency matrix denoted as $\hat{A}$, we map nodes to token representations and set weights of the edge between node $i$ and node $j$ to $\exp(Q_i^\top K_j)$. Then $(i, j)$ entry of a normalized adjacency matrix is $\tilde{A}_{ij} = \hat{A}_{ij}/\hat{D}_{ij} = \exp(Q_i^\top K_j)/\sum_k \exp(Q_i^\top K_k)$, where diagonal matrix $\hat{D}_{ii} = \sum_j \hat{A}_{ij}$. Apparently, $\tilde{A}$ is equal to the form of self-attention matrix defined in Transformers. Simultaneously, $\tilde{A}$ plays a major part in the message-passing scheme by deciding which nodes to exchange information with.

**Oversmoothing in GNNs and Transformers.** The term oversmoothing is firstly proposed by Li et al. (2018) in research of GNNs. Intuitively, representations converge to a constant after repeatedly exchanging messages with neighbors when the layer goes to infinity. Zhou et al. (2020) mathematically proves that, under some conditions, the convergence point carries only information of the graph topology. Coincidentally, an oversmoothing-like phenomenon is also observed in Transformers. Unlike CNNs, Transformers can not benefit from simply deepening layers, and even saturates with depth increasing. Early works empirically ascribe it to attention/feature collapse or patch/token uniformity (Tang et al., 2021; Zhou et al., 2021; Gong et al., 2021; Yan et al., 2022). To be specific, the attention maps tend to be overly similar in later layers, whereby features insufficiently exchange information and lose diversity in the end. Outputs of pure transformers, i.e., attention without skip connections or MLPs, are also observed to converge to a rank-1 matrix (Dong et al., 2021). For illustration proposes, we also refer to the degeneration issue in Transformers as oversmoothing.

**Whitening Methods for Dimensional Collapse.** Whitening methods ensure that the covariance matrix of normalized outputs is diagonal, making dimensions mutually independent and implicitly solving dimensional collapse. Huang et al. (2018; 2019); Siarohin et al. (2018); Ermolov et al. (2021) all adopt the idea of whitening, but differ in calculating details of whitening matrix and application domain. Compared with them, we avoid complicated calculations on the squared root of the inverse covariance matrix and delicate design of backward pass for differentiability. Moreover, Huang et al. (2018; 2019) are proposed for convolution operations, Siarohin et al. (2018) is for GAN, and Ermolov et al. (2021) works in self-supervised learning. In contrast, we borrow the idea from contrastive learning and solve oversmoothing in completely different fields like Transformers and GNNs.

## 3 MITIGATING OVERSMOOTHING FROM THE PERSPECTIVE OF CONTRASTIVE LEARNING

In this section, we first empirically demonstrate that current similarity metric fails to characterize dimensional collapse, thus overlooking a crucial part of oversmoothing. To address this problem, we draw inspiration from contrastive learning whose uniformity property naturally rules out dimensional collapse. Specifically, we transfer the uniformity loss to a loss that directly acts on representations. By unfolding the optimization steps along this loss, we induce a normalization layer called ContraNorm. Theoretically, we prove our proposed layer helps mitigate dimensional collapse.

### 3.1 THE CHARACTERIZATION OF OVERSMOOTHING

In this part, we begin by highlighting the limitations of existing metrics in characterizing oversmoothing. These limitations motivate us to adopt the effective rank metric, which has been demonstrated to be effective in capturing the degree of dimensional collapse in contrastive learning.

Taking the oversmoothing problem of Transformers as an example without loss of generality, a prevailing metric is evaluating attention map similarity (Wang et al., 2022; Gong et al., 2021; Shi et al., 2022). The intuition is that as attention map similarity converges to one, feature similarity increases, which can result in a loss of representation expressiveness and decreased performance. However, by conducting experiments on transformer structured models like ViT and BERT, we find

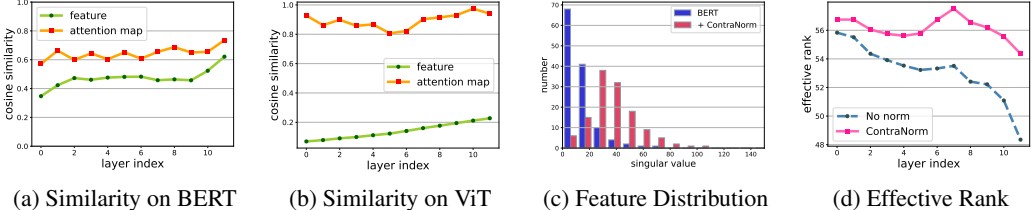

| (a) Similarity on BERT | (b) Similarity on ViT | (c) Feature Distribution | (d) Effective Rank |

Figure 2: Comparison between different metrics of oversmoothing of 12-layer BERT on GLUE sub-task STS-B and 12-layer ViT on CIFAR10. Fig. 2a and Fig. 2b show feature and attention map cosine similarity along layer index. Fig. 2c and Fig. 2d show the singular value distribution of features and effective rank of BERT w/ and w/o ContraNorm.

that high attention map similarity does not necessarily correspond to high feature similarity. As shown in Fig. 2a and Fig. 2b, although attention map similarity is close to 0.8 or even higher, the feature similarity remains below 0.5 in most cases. It means the oversmoothing problem still occurs even with a low feature similarity. This finding suggests that similarity can not fully depict the quality of representations and the oversmoothing problem.

Intuitively, we can consider a case that representations in the latent embedding space do not shrink to one single point but span a low dimensional space. In such cases, feature similarity may be relatively low, but representations still lose expressive power. Such representation degeneration problem is known as *dimensional collapse*, and it is widely discussed in the literature of contrastive learning. In contrastive learning, common practice to describe dimensional collapse is the vanishing distribution of singular values (Gao et al., 2019; Ethayarajh, 2019; Jing et al., 2022). To investigate whether dimensional collapse occurs in Transformers, we draw the singular value distribution of features in the last block of 12-layer BERT. As shown in Fig. 2c, the insignificant (nearly zero) values dominate singular value distribution in the deep layer of vanilla BERT, indicating that representations reside on a low-dimensional manifold and dimensional collapse happens. To show the collapse tendency along layer index, we simplify the singular value distribution to a concise scalar *effective rank* (erank) (Roy & Vetterli, 2007), which covers the full singular value spectrum.

**Definition 3.1** (Effective Rank). *Considering matrix $\boldsymbol{X} \in \mathbb{R}^{m \times n}$ whose singular value decomposition is given by $\boldsymbol{X} = \boldsymbol{U}\boldsymbol{\Sigma}\boldsymbol{V}$, where $\boldsymbol{\Sigma}$ is a diagonal matrix with singular values $\sigma_1 \geq \sigma_2 \geq \cdots \geq \sigma_Q \geq 0$ with $Q = \min\{m, n\}$. The distribution of singular values is defined as $L_1$-normalized form $p_i = \sigma_i / \sum_{k=1}^Q |\sigma_k|$. The effective rank of the matrix $\boldsymbol{X}$, denoted as erank($\boldsymbol{X}$), is defined as erank($\boldsymbol{X}$) = $\exp(H\{p_1, p_2, \cdots, p_Q\})$, where $H(p_1, p_2, \cdots, p_Q)$ is the Shannon entropy given by $H(p_1, p_2, \cdots, p_Q) = -\sum_{k=1}^Q p_k \log p_k$.*

Based on erank, we revisit the oversmoothing issue of Transformers in Fig. 2d. We can see that the effective rank descends along with layer index, indicating an increasingly imbalanced singular value distribution in deeper layers. This finding not only verifies dimensional collapse does occur in Transformers, but also indicates the effectiveness of using effective rank as a metric to detect this issue.

### 3.2 INSPIRATIONS FROM THE UNIFORMITY LOSS IN CONTRASTIVE LEARNING

The core idea of contrastive learning is maximizing agreement between augmented views of the same example (i.e., positive pairs) and disagreement of views from different samples (i.e. negative pairs). A popular form of contrastive learning optimizes feature representations using a loss function with limited negative samples (Chen et al., 2020). Concretely, given a batch of randomly sampled examples, for each example we generate its augmented positive views and finally get a total of $N$ samples. Considering an encoder function $f$, the contrastive loss for the positive pair $(i, i^+)$ is

$$\mathcal{L}(i, i^+) = -\log \frac{\exp(f(\boldsymbol{x}_i)^\top f(\boldsymbol{x}_{i^+})/\tau)}{\sum_{k=1}^N \exp(f(\boldsymbol{x}_i)^\top f(\boldsymbol{x}_k)/\tau) - \exp(f(\boldsymbol{x}_i)^\top f(\boldsymbol{x}_i)/\tau)}, \quad (1)$$

where $\tau$ denotes a temperature factor. The loss can be decoupled into two parts named *alignment loss* and *uniformity loss*:

$$\mathcal{L}_{\text{align}}(i, i^+) = -f(\boldsymbol{x}_i)^\top f(\boldsymbol{x}_{i+})/\tau \qquad \mathcal{L}_{\text{uniform}}(i) = \log \sum_{k=1}^N \exp(f(\boldsymbol{x}_i)^\top f(\boldsymbol{x}_k)/\tau). \qquad (2)$$

The alignment loss encourages feature representations for positive pairs to be similar, thus being invariant to unnecessary noises. However, training with only the alignment loss will result in a trivial solution where all representations are identical. In other words, complete collapse happens. While batch normalization (Ioffe & Szegedy, 2015) can help avoid this issue, it cannot fully prevent the problem of dimensional collapse, which still negatively impacts learning (Hua et al., 2021).

Thanks to the property of uniformity loss, dimensional collapse can be solved effectively. Reviewing the form of uniformity loss in Eq.(2), it maximizes average distances between all samples, resulting in embeddings that are roughly uniformly distributed in the latent space, and thus more information is preserved. The inclusion of the uniformity loss in the training process helps to alleviate dimensional collapse. Intuitively, it can also serve as a sharp tool for alleviating oversmoothing in models such as GNNs and Transformers.

An alternative approach to address oversmoothing is to directly incorporate the uniformity loss into the training objective. However, our experiments reveal that this method has limited effectiveness (see Appendix D for more details). Instead, we propose a normalization layer that can be easily integrated into various models. Our approach utilizes the uniformity loss as an underlying energy function of the proposed layer, such that a descent step along the energy function corresponds to a forward pass of the layer. Alternatively, we can view the layer as the unfolded iterations of an optimization function. This perspective is adopted to elucidate GNNs (Yang et al., 2021; Zhu et al., 2021), Transformers (Yang et al., 2022), and classic MLPs (Xie et al., 2021).

Note that the uniformity loss works by optimizing model parameters while what the normalization layer directly updates is representation itself. So we first transfer the uniformity loss, which serves as a training loss, to a kind of architecture loss. Consider a fully connected graph with restricted number of nodes, where node $\boldsymbol{h}_i$ is viewed as representation of a random sample $\boldsymbol{x}_i$. Reminiscent of $\mathcal{L}_{\text{uniform}}$, we define $\hat{\mathcal{L}}_{\text{uniform}}$ over all nodes in the graph as

$$\hat{\mathcal{L}}_{\text{uniform}} = \sum_i \mathcal{L}_{\text{uniform}}(i) = \sum_i \log \sum_j e^{\boldsymbol{h}_i^\top \boldsymbol{h}_j/\tau}. \qquad (3)$$

This form of uniformity loss is defined directly on representations, and we later use it as the underlying energy function for representation update.

### 3.3 THE PROPOSED CONTRANORM

Till now, we are able to build a connection between layer design and the unfolded iterations of descent steps used to minimize uniformity loss. Specifically, we take the derivative of $\hat{L}_{\text{uniform}}$ on node representations:

$$\frac{\partial \hat{\mathcal{L}}_{\text{uniform}}}{\partial \boldsymbol{h}_i} = \sum_j \frac{\exp(\boldsymbol{h}_i^\top \boldsymbol{h}_j/\tau)}{\sum_k \exp(\boldsymbol{h}_i^\top \boldsymbol{h}_k/\tau)} \boldsymbol{h}_j/\tau + \sum_j \frac{\exp(\boldsymbol{h}_i^\top \boldsymbol{h}_j/\tau)}{\sum_k \exp(\boldsymbol{h}_j^\top \boldsymbol{h}_k/\tau)} \boldsymbol{h}_j/\tau, \qquad (4)$$

by denoting feature matrix as $\boldsymbol{H}$ with $\boldsymbol{h}_i^\top$ as the $i$-th row, we rewrite Eq.(4) into a matrix form for simplicity:

$$\frac{\partial \hat{\mathcal{L}}_{\text{uniform}}}{\partial \boldsymbol{H}} = (\boldsymbol{D}^{-1}\boldsymbol{A} + \boldsymbol{A}\boldsymbol{D}^{-1})\boldsymbol{H}/\tau, \qquad (5)$$

where $\boldsymbol{A} = \exp(\boldsymbol{H}\boldsymbol{H}^\top/\tau)$ and $\boldsymbol{D} = deg(\boldsymbol{A})^1$. In order to reduce the uniformity loss $\hat{\mathcal{L}}_{\text{uniform}}$, a natural way is to take a single step performing gradient descent, which is to update representations $\boldsymbol{H}$ in the way of

$$\boldsymbol{H}_t = \boldsymbol{H}_b - s \times \frac{\partial \hat{\mathcal{L}}_{\text{uniform}}}{\partial \boldsymbol{H}_b} = \boldsymbol{H}_b - s/\tau \times (\boldsymbol{D}^{-1}\boldsymbol{A} + \boldsymbol{A}\boldsymbol{D}^{-1})\boldsymbol{H}_b, \qquad (6)$$

---

[1]$deg(\boldsymbol{A})$ is a diagonal matrix, whose element in the $i$-th row and the $i$-th column is the sum of the $i$-th row of $\boldsymbol{A}$.

where $\boldsymbol{H}_b$ and $\boldsymbol{H}_t$ denote the representations before and after the update, respectively, and $s$ is the step size of the gradient descent. By taking this update after a certain representation layer, we can reduce the uniformity loss of the representations and thus help ease the dimensional collapse.

In Eq.(6), there exist two terms $\boldsymbol{D}^{-1}\boldsymbol{A}$ and $\boldsymbol{A}\boldsymbol{D}^{-1}$ multiplied with $\boldsymbol{H}_b$. Empirically, the two terms play a similar role in our method. Note that the first term is related to self-attention matrix in Transformers, so we only preserve it and discard the second one. Then Eq.(6) becomes

$$\boldsymbol{H}_t = \boldsymbol{H}_b - s/\tau \times (\boldsymbol{D}^{-1}\boldsymbol{A})\boldsymbol{H}_b = \boldsymbol{H}_b - s/\tau \times \mathrm{softmax}(\boldsymbol{H}_b\boldsymbol{H}_b^\top)\boldsymbol{H}_b. \tag{7}$$

In fact, the operation corresponds to the stop-gradient technique, which is widely used in contrastive learning methods (He et al., 2020; Grill et al., 2020; Tao et al., 2022). By throwing away some terms in the gradient, stop-gradient makes the training process asymmetric and thus avoids representation collapse with less computational overhead, which is discussed in detail in Appendix A.

However, the layer induced by Eq.(7) still can not ensure uniformity on representations. Consider an extreme case where $\mathrm{softmax}(\boldsymbol{H}_b\boldsymbol{H}_b^\top)$ is equal to identity matrix $\boldsymbol{I}$. Eq.(7) becomes $\boldsymbol{H}_t = \boldsymbol{H}_b - s/\tau \times \boldsymbol{H}_b = (1 - s/\tau)\boldsymbol{H}_b$, which just makes the scale of $\boldsymbol{H}$ smaller and does not help alleviate the complete collapse. To keep the representation stable, we propose two methods:

**Regularization.** We add a regularization term $-\frac{1}{2}\sum_i \|h_i\|_2^2$ to the uniformity loss. When the regularization term becomes smaller, the norm of $h_i$ becomes larger. Therefore, adding this term can help prevent the norm of representation $h_i$ from becoming smaller. In this way, the update form becomes

$$\boldsymbol{H}_t = (1 + s)\boldsymbol{H}_b - s/\tau \times \mathrm{softmax}(\boldsymbol{H}_b\boldsymbol{H}_b^\top)\boldsymbol{H}_b. \tag{8}$$

**Proposition 1.** *Let $\boldsymbol{e} = (1, 1, \ldots, 1)^\top/\sqrt{n}$. For attention matrix $\bar{\boldsymbol{A}} = \mathrm{softmax}(\boldsymbol{H}_b\boldsymbol{H}_b^\top)$, let $\sigma_{\min}$ be the smallest eigenvalue of $\boldsymbol{P} = (\boldsymbol{I} - \boldsymbol{e}\boldsymbol{e}^\top)(\boldsymbol{I} - \bar{\boldsymbol{A}}) + (\boldsymbol{I} - \bar{\boldsymbol{A}})^\top(\boldsymbol{I} - \boldsymbol{e}\boldsymbol{e}^\top)$. For the update in Eq.8, i.e. $\boldsymbol{H}_t = ((1 + s)\boldsymbol{I} - s\bar{\boldsymbol{A}})\boldsymbol{H}_b,\ s > 0$, we have $Var(\boldsymbol{H}_t) \geq (1 + s\sigma_{\min}) \cdot Var(\boldsymbol{H}_b)$. Especially, if $\sigma_{\min} \geq 0$, we have $Var(\boldsymbol{H}_t) \geq Var(\boldsymbol{H}_b)$.*

Proposition 1 gives a bound for the ratio of the variance after and before the update. It shows that the change of the variance is influenced by the symmetric matrix $\boldsymbol{P} = (\boldsymbol{I} - \boldsymbol{e}\boldsymbol{e}^\top)(\boldsymbol{I} - \bar{\boldsymbol{A}}) + (\boldsymbol{I} - \bar{\boldsymbol{A}})^\top(\boldsymbol{I} - \boldsymbol{e}\boldsymbol{e}^\top)$. If $\boldsymbol{P}$ is a semi-positive definite matrix, we will get the result that $Var(\boldsymbol{H}_t) \geq Var(\boldsymbol{H}_b)$, which indicates that the representations become more uniform. In Appendix B, we will give out some sufficient conditions for that $\boldsymbol{P}$ is semi-positive definite.

**LayerNorm.** We leverage layer normalization (Ba et al., 2016), which adjusts the representation according to its mean and variance. The update form of the layer normalization on representation $\boldsymbol{h}_i$ is $\mathrm{LayerNorm}(\boldsymbol{h}_i) = \gamma \cdot ((\boldsymbol{h}_i - \mathrm{mean}(\boldsymbol{h}_i))/\sqrt{\mathrm{Var}(\boldsymbol{h}_i) + \varepsilon}) + \beta$, where $\gamma$ and $\beta$ are learnable parameters and $\varepsilon = 10^{-5}$ prevents the denominator from becoming zero. The learnable parameters $\gamma$ and $\beta$ can rescale the representation $\boldsymbol{h}_i$ to help ease the problem. We append the layer normalization to Eq.(7) as

$$\boldsymbol{H}_t = \mathrm{LayerNorm}\left(\boldsymbol{H}_b - s/\tau \times \mathrm{softmax}(\boldsymbol{H}_b\boldsymbol{H}_b^\top)\boldsymbol{H}_b\right), \tag{9}$$

where applying the layer normalization to a representation matrix $\boldsymbol{H}$ means applying the layer normalization to all its components $\boldsymbol{h}_1, \ldots, \boldsymbol{h}_n$. We empirically compare the two proposed methods and find their performance comparable, while the second one performs slightly better. Therefore, we adopt the second update form and name it Contrastive Normalization (**ContraNorm**). The ContraNorm layer can be added after any representation layer to reduce the uniformity loss and help relieve the dimensional collapse. We discuss the best place to plug our ContraNorm in Appendix E. Given $\mathbf{H}_b \in \mathbb{R}^{n \times d}$ where $n$ and $d$ is the number of samples in the batch and hidden dimension, the time complexity of ContraNorm is $\mathcal{O}(n^2 d)$, which is the same order as the self-attention operation in Transformer. For enhancing scalability, we propose a modified version of ContraNorm with a linear complexity in the number of samples. See Appendix F for more details.

## 3.4 THEORETICAL ANALYSIS

In this part, we will give a theoretical result on our proposed ContraNorm method. We will show that with a different form of the uniformity loss, the ContraNorm update can help alleviate the dimensional collapse.

**Proposition 2.** *Consider the update form*

$$\boldsymbol{H}_t = (1 + s)\boldsymbol{H}_b - s(\boldsymbol{H}_b\boldsymbol{H}_b^\top)\boldsymbol{H}_b, \tag{10}$$

*let $\sigma_{\max}$ be the largest singular value of $\boldsymbol{H}_b$. For $s > 0$ satisfying $1 + (1 - \sigma_{\max}^2)s > 0$, we have* $\mathrm{erank}(\boldsymbol{H}_t) > \mathrm{erank}(\boldsymbol{H}_b)$.

Proposition 2 gives a promise of alleviating dimensional collapse under a special update form. Denote $\hat{\mathcal{L}}_{\mathrm{dim}} = tr((\boldsymbol{I} - \boldsymbol{H}\boldsymbol{H}^\top)^2)/4$, then $\hat{\mathcal{L}}_{\mathrm{dim}}$ shares a similar form with Barlow Twins (Zbontar et al., 2021). This loss tries to equate the diagonal elements of the similarity matrix $\boldsymbol{H}\boldsymbol{H}^\top$ to 1 and equate the off-diagonal elements of the matrix to 0, which drives different features becoming orthogonal, thus helping the features become more uniform in the space. Note that $\frac{\partial \hat{\mathcal{L}}_{\mathrm{dim}}}{\partial \boldsymbol{H}} = (\boldsymbol{I} - \boldsymbol{H}\boldsymbol{H}^\top)\boldsymbol{H}$, therefore, Eq.(10) can be rewritten as $\boldsymbol{H}_t = \boldsymbol{H}_b + s\frac{\partial \hat{\mathcal{L}}_{\mathrm{dim}}(\boldsymbol{H}_b)}{\partial \boldsymbol{H}_b}$, which implies that this update form is in fact a ContraNorm with a different uniformity loss. Proposition 2 claims that this update can increase the effective rank of the representation matrix, when $s$ satisfies $1 + (1 - \sigma_{\max})s > 0$. Note that no matter whether $\sigma_{\max} > 0$ or not, if $s$ is sufficiently close to 0, the condition will be satisfied. Under this situation, the update will alleviate the dimensional collapse.

## 4 EXPERIMENTS

In this section, we demonstrate the effectiveness of ContraNorm by experiments including 1) language understanding tasks on GLUE datasets with BERT and ALBERT (Lan et al., 2020) as the backbones; 2) image classification task on ImageNet100 and ImageNet1k datasets with DeiT as the base model; 3) fully supervised graph node classification task on popular graph datasets with GCN as the backbone. Moreover, we conduct ablative studies on ContraNorm variants comparison and hyperparameters sensitivity analysis.

### 4.1 EXPERIMENTS ON LANGUAGE MODELS

To corroborate the potential of ContraNorm, we integrate it into two transformer structured models: BERT and ALBERT, and evaluate it on GLUE datasets. GLUE includes three categories of tasks: (i) single-sentence tasks CoLA and SST-2; (ii) similarity and paraphrase tasks MRPC, QQP, and STS-B; (iii) inference tasks MNLI, QNLI, and RTE. For MNLI task, we experiment on both the matched (MNLI-m) and mismatched (MNLI-mm) versions.

**Setup.** We default plug ContraNorm after the self-attention module and residual connection (more position choices are in Appendix E). We use a batch size of 32 and fine-tune for 5 epochs over the data for all GLUE tasks. For each task, we select the best scale factor $s$ in Eq.(6) among $(0.005, 0.01, 0.05, 0.1, 0.2)$. We use base models (BERT-base and ALBERT-base) of 12 stacked blocks with hyperparameters fixed for all tasks: number of hidden size 128, number of attention heads 12, maximum sequence length 384. We use Adam (Kingma & Ba, 2014) optimizer with the learning rate of $2e - 5$.

**Results.** As shown in Table 1, ContraNorm substantially improves results on all datasets compared with vanilla BERT. Specifically, our ContraNorm improves the previous average performance from 82.59% to 83.39% for BERT backbone and from 83.74% to 84.67% for ALBERT backbone. We also submit our trained model to GLUE benchmark leaderboard and the results can be found in Appendix G. It is observed that BERT with ContraNorm also outperforms vanilla model across all datasets. To verify the de-oversmoothing effect of ContraNorm. We build models with/without ContraNorm on various layer depth settings. The performance comparison is shown in Fig. 3a. Constant stack of blocks causes obvious deterioration in vanilla BERT, while BERT with ContraNorm maintains competitive advantage. Moreover, for deep models, we also show the tendency of variance and effective rank in Fig. 3b and Fig. 3c, which verifies the power of ContraNorm in alleviating complete collapse and dimensional collapse, respectively.

### 4.2 EXPERIMENTS ON VISION TRANSFORMERS

We also validate effectiveness of ContraNorm in computer vision field. We choose DeiT as backbone and models are trained from scratch. Experiments with different depth settings are evaluated on ImageNet100 and ImageNet1k datasets. Based on the Timm (Wightman, 2019) and DeiT repositories, we insert ContraNorm into base model intermediately following self-attention module.

**Setup.** We follow the training recipe of Touvron et al. (2021), and make minimal changes to hyperparameters. Specifically, we use AdamW (Loshchilov & Hutter, 2019) optimizer with cosine learning rate decay. We train each model for 150 epochs and the batch size is set to 1024. Aug-

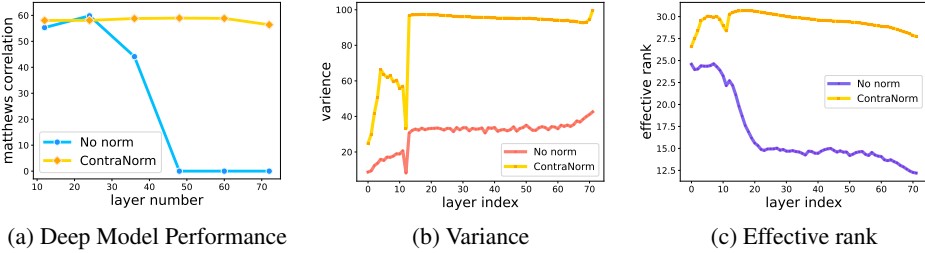

(a) Deep Model Performance      (b) Variance      (c) Effective rank

Figure 3: Comparison of BERT and BERT+ContraNorm on COLA with various layer depths. Fig.(3a) presents performances of models with block number=12, 24, 36, 48, 60, 72. Fig.(3b) and Fig.(3c) show the tendency of features variance and effective rank for models with 72 blocks.

Table 1: Results comparison on validation set of GLUE tasks. Following (Devlin et al., 2018), we report F1 scores for QQP and MRPC, Spearman correlations for STS-B, and accuracy scores for the other tasks. ContraNorm* denotes the best model when varying plugging positions of ContraNorm. **Avg** denotes the average performance on all the tasks and bold denotes the best performance.

| Dataset | COLA | SST-2 | MRPC | QQP | STS-B | MNLI-m | MNLI-mm | QNLI | RTE | Avg |
|---|---|---|---|---|---|---|---|---|---|---|
| BERT-base | 55.28 | 92.89 | 88.96 | 88.24 | 88.61 | 84.65 | 84.61 | 91.51 | 68.59 | 82.59 |
| BERT + ContraNorm | 58.83 | 93.12 | 89.49 | **88.30** | 88.66 | **84.87** | 84.66 | **91.78** | 70.76 | 83.39 |
| BERT + ContraNorm* | **59.57** | **93.23** | **89.97** | **88.30** | **88.93** | **84.87** | **84.67** | **91.78** | **70.76** | **83.54** |
| ALBERT-base | 57.35 | **93.69** | 92.09 | 87.23 | 90.54 | 84.56 | 84.37 | 90.90 | 76.53 | 83.74 |
| ALBERT + ContraNorm | 58.51 | 92.89 | 92.86 | 87.45 | 90.56 | 84.33 | 84.62 | 91.76 | **79.06** | 84.67 |
| ALBERT + ContraNorm* | **58.76** | 93.23 | **92.89** | **87.67** | **90.72** | **84.69** | **84.95** | **92.28** | **79.06** | **84.92** |

Table 2: Test accuracy (%) comparison results. For ImageNet100 and ImageNet1k, we use DeiT-tiny and DeiT-small as the baseline respectively. The block number is set to 12, 16 and 24. The best result for each dataset is bolded.

| Dataset | Model | #L=12 | #L=16 | #L=24 |
|---|---|---|---|---|
| ImageNet100 | DeiT-tiny | 76.58 | 75.34 | 76.76 |
|  | +ContraNorm | **79.34** | **80.44** | **81.28** |
| ImageNet1k | DeiT-small | 77.32 | 78.25 | 77.69 |
|  | +ContraNorm | **77.80** | **79.04** | **78.67** |

mentation techniques are used to boost the performance. For all experiments, the image size is set to be $224 \times 224$. For Imagenet100, we set the scale factor $s$ to 0.2 for all layer depth settings. For Imagenet1k, we set $s$ to 0.2 for models with 12 and 16 blocks, and raise it to 0.3 for models with 24 blocks.

**Results.** In Table 2, DeiT with ContraNorm outperforms vanilla DeiT with all layer settings on both datasets. Typically, our method shows a gain of nearly 5% on test accuracy for ImageNet100. For ImageNet1k, we boost the performance of DeiT with 24 blocks from 77.69% to 78.67%.

### 4.3 EXPERIMENTS ON GRAPH NEURAL NETWORKS

We implement ContraNorm as a simple normalization layer after each graph convolution of GCN, and evaluate it on fully supervised graph node classification task. For datasets, we choose two popular citation networks Cora (McCallum et al., 2000) and Citeseer (Giles et al., 1998), and Wikipedia article networks Chameleon and Squirrel (Rozemberczki et al., 2021). More information of datasets is deferred to Appendix H. We compare ContraNorm against a popular normalization layer PairNorm (Zhao & Akoglu, 2020) designed for preventing oversmoothing. We also take LayerNorm as a baseline by setting the scale $s = 0$.

**Setup.** We follow data split setting in Kipf & Welling (2017) with train/validation/test splits of 60%, 20%, 20%, respectively. To keep fair in comparison, we fix the hidden dimension to 32, and dropout rate to 0.6 as reported in Zhao & Akoglu (2020). We choose the best of scale controller $s$ in range of $\{0.2, 0.5, 0.8, 1.0\}$ for both PairNorm and ContraNorm. For PairNorm, we choose the best variant presented by Zhao & Akoglu (2020).

**Results.** As shown in Table 3, in shallow layers (e.g., two layer), the addition of ContraNorm reduces the accuracy of vanilla GCN by a small margin, while it helps prevent the performance from sharply deteriorating as the layer goes deeper. ContraNorm outperforms PairNorm and Lay-

Table 3: Test accuracy (%) comparison results. We use GCN as backbone and apply LayerNorm, PairNorm and ContraNorm, respectively. We fairly tune the scale parameter in LayerNorm and ContraNorm. The layer number is set to 2, 4, 8, 16, 32. For every layer setting, the best accuracy is marked in blue background, and the second best is underlined. Results are averaged over 5 runs.

| Dataset | Model | #L=2 | #L=4 | #L=8 | #L=16 | #L=32 |
|---|---|---|---|---|---|---|
| Cora | Vanilla GCN | $81.75 \pm 0.51$ | $72.61 \pm 2.42$ | $17.71 \pm 6.89$ | $20.71 \pm 8.54$ | $19.69 \pm 9.54$ |
| | +LayerNorm | $79.96 \pm 0.73$ | $77.45 \pm 0.67$ | $39.09 \pm 4.68$ | $7.79 \pm 0.00$ | $7.79 \pm 0.00$ |
| | +PairNorm | $75.32 \pm 1.05$ | $72.64 \pm 2.67$ | $71.86 \pm 3.31$ | $54.11 \pm 9.49$ | $36.62 \pm 2.73$ |
| | +ContraNorm | $79.75 \pm 0.33$ | $77.02 \pm 0.96$ | $74.01 \pm 0.64$ | $68.75 \pm 2.10$ | $46.39 \pm 2.46$ |
| CiteSeer | Vanilla GCN | $69.18 \pm 0.34$ | $55.01 \pm 4.36$ | $19.65 \pm 0.00$ | $19.65 \pm 0.00$ | $19.65 \pm 0.00$ |
| | +LayerNorm | $63.27 \pm 1.15$ | $60.91 \pm 0.76$ | $33.74 \pm 6.15$ | $19.65 \pm 0.00$ | $19.65 \pm 0.00$ |
| | +PairNorm | $61.59 \pm 1.35$ | $53.01 \pm 2.19$ | $55.76 \pm 4.45$ | $44.21 \pm 1.73$ | $36.68 \pm 2.55$ |
| | +ContraNorm | $64.06 \pm 0.85$ | $60.55 \pm 0.72$ | $59.30 \pm 0.67$ | $49.01 \pm 3.49$ | $36.94 \pm 1.70$ |
| Chameleon | Vanilla GCN | $45.79 \pm 1.20$ | $37.85 \pm 1.35$ | $22.37 \pm 0.00$ | $22.37 \pm 0.00$ | $23.37 \pm 0.00$ |
| | +LayerNorm | $63.95 \pm 1.29$ | $55.79 \pm 1.25$ | $34.08 \pm 2.62$ | $22.37 \pm 0.00$ | $22.37 \pm 0.00$ |
| | +PairNorm | $62.24 \pm 1.73$ | $58.38 \pm 1.48$ | $49.12 \pm 2.32$ | $37.54 \pm 1.70$ | $30.66 \pm 1.58$ |
| | +ContraNorm | $64.78 \pm 1.68$ | $58.73 \pm 1.12$ | $48.99 \pm 1.52$ | $40.92 \pm 1.74$ | $35.44 \pm 3.16$ |
| Squirrel | Vanilla GCN | $29.47 \pm 0.96$ | $19.31 \pm 0.00$ | $19.31 \pm 0.00$ | $19.31 \pm 0.00$ | $19.31 \pm 0.00$ |
| | +LayerNorm | $43.04 \pm 1.25$ | $29.64 \pm 5.50$ | $19.63 \pm 0.45$ | $19.96 \pm 0.44$ | $19.40 \pm 0.19$ |
| | +PairNorm | $43.86 \pm 0.41$ | $40.25 \pm 0.55$ | $36.03 \pm 1.43$ | $29.55 \pm 2.19$ | $29.05 \pm 0.91$ |
| | +ContraNorm | $47.24 \pm 0.66$ | $40.31 \pm 0.74$ | $35.85 \pm 1.58$ | $32.37 \pm 0.93$ | $27.80 \pm 0.72$ |

Table 4: Performance comparison among different variants of ContraNorm. SG, LN, $L_2$N are the abbreviations of stop gradient, layer normalization and $L_2$ normalization, respectively. All experiments are conducted on GLUE tasks with the same parameter settings. **Avg** denotes the average performance on all the tasks. We bold the best result for each task.

| Variants | | | Datasets | | | | | | | | | Avg |
|---|---|---|---|---|---|---|---|---|---|---|---|---|
| SG | LN | $L_2$N | COLA | SST-2 | MRPC | QQP | STS-B | MNLI-m | MNLI-mm | QNLI | RTE | |
| | ✓ | | 58.80 | **93.12** | 89.60 | 88.35 | **88.97** | 84.81 | **84.67** | 91.47 | 68.23 | 83.11 |
| ✓ | | ✓ | **59.82** | 93.00 | 89.64 | **88.37** | 88.92 | 84.72 | 84.58 | **91.58** | **68.95** | 83.29 |
| ✓ | ✓ | | 59.57 | **93.12** | **89.97** | 88.30 | 88.93 | **84.84** | **84.67** | **91.58** | **68.95** | **83.33** |

erNorm in the power of de-oversmoothing. Here, we show the staple in diluting oversmoothing is ContraNorm, and LayerNorm alone fails to prevent GCN from oversmoothing, even amplifying the negative effect on Cora with more than 16 layers.

## 4.4 ABLATION STUDY

**Comparison of different ContraNorm variants.** Recalling Section 3.3, we improve ContraNorm with stop-gradient technique (SG) by masking the second term. For solving data representation instability, we apply layer normalization (LN) to the original version, while for the convenience of theoretical analysis, layer normalization is replaced by $L_2$ normalization ($L_2$N). Here, we investigate the effect of these tricks and the results are shown in Table 4. Compared with the variant with only LayerNorm, ContraNorm with both stop gradient and layer normalization presents better performance. As for the two normalization methods, they are almost comparable to our methods, which verifies the applicability of our theoretical analysis.

**Hyperparameters sensitivity analysis.** We conduct an ablation study regarding the gains of ContraNorm while using different values of scale factor $s$. In Appendix I, we show that ContraNorm is robust in an appropriate range of $s$.

## 5 CONCLUSION

In this paper, we point out the deficiencies of current metrics in characterizing over-smoothing, and highlight the importance of taking dimensional collapse as the entry point for oversmoothing analysis. Inspired by the uniformity loss in contrastive learning, we design an optimization-induced normalization layer ContraNorm. Our theoretical findings indicate ContraNorm mitigates dimensional collapse successfully by reducing variance and effective rank of representations. Experiments show that ContraNorm boosts the performance of Transformers and GNNs on various tasks. Our work provides a new perspective from contrastive learning on solving the oversmoothing problem, which helps motivate the following extensive research.

## ACKNOWLEDGEMENT

Yisen Wang is partially supported by the National Key R&D Program of China (2022ZD0160304), the National Natural Science Foundation of China (62006153), Open Research Projects of Zhejiang Lab (No. 2022RC0AB05), and Huawei Technologies Inc. Xiaojun Guo thanks for Hanqi Yan for her kind guidance on the implementation of BERT models. We thank anonymous reviewers for their constructive advice on this work.

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

## A  ILLUSTRATION ON THE STOP-GRADIENT OPERATION IN EQ.(7)

In this section, we will illustrate the stop-gradient operation in Eq.(7) by using the framework proposed by (Tao et al., 2022). The original update form should be Eq.(6):

$$\boldsymbol{H}_t = \boldsymbol{H}_b - s \times \frac{\partial \hat{\mathcal{L}}_{\text{uniform}}}{\partial \boldsymbol{H}_b} = \boldsymbol{H}_b - s/\tau \times (\boldsymbol{D}^{-1}\boldsymbol{A} + \boldsymbol{A}\boldsymbol{D}^{-1})\boldsymbol{H}_b.$$

We take SimCLR(Chen et al., 2020) as the contrastive learning framework. Tao et al. (2022) have studied the stop-gradient form of SimCLR and illustrated that the stop-gradient operation will make a similar performance with the original one. Based on this, we will elaborate on how $\boldsymbol{A}\boldsymbol{D}^{-1}$ is removed in the following part. In fact, we can directly illustrate how the second term in Eq(4) can be omitted.

$$\frac{\partial \hat{\mathcal{L}}_{\text{uniform}}}{\partial \boldsymbol{h}_i} = \sum_j \frac{\exp(\boldsymbol{h}_i^\top \boldsymbol{h}_j/\tau)}{\sum_k \exp(\boldsymbol{h}_i^\top \boldsymbol{h}_k/\tau)}\boldsymbol{h}_j/\tau + \sum_j \frac{\exp(\boldsymbol{h}_i^\top \boldsymbol{h}_j/\tau)}{\sum_k \exp(\boldsymbol{h}_j^\top \boldsymbol{h}_k/\tau)}\boldsymbol{h}_j/\tau.$$

In SimCLR, by denoting the normalized features from the online branch as $u_i^o, i = 1, 2, \ldots, n$, the normalized features from the target branch (although the two branches have no differences in SimCLR) are $u_i^t, i = 1, 2, \ldots, n$ and $\mathcal{V} = \{u_1^o, u_2^o, \ldots, u_n^o, u_1^t, u_2^t, \ldots, u_n^t\}$, the SimCLR loss can be represented as

$$L = -\frac{1}{2n} \sum_{u_1 \in \mathcal{V}} \log \frac{\exp(u_1^\top u_1'/\tau)}{\sum_{u \in \mathcal{V}/u_1} \exp(u_1^\top u/\tau)}$$

where $u_1$ and $u_1'$ are positive pairs and $\tau$ is the temperature. Then the gradient of $L$ on $u_1^o$ can be calculated as

$$\frac{\partial L}{\partial u_1^o} = \frac{1}{2\tau n}\left(-u_1^t + \sum_{v \in \mathcal{V}/u_1^o} s_v v\right) + \frac{1}{2\tau n}\left(-u_1^t + \sum_{v \in \mathcal{V}/u_1^o} t_v v\right)$$

where

$$s_v = \frac{\exp(u_1^{o\top} v/\tau)}{\sum_{v' \in \mathcal{V}/u_1^o} \exp(u_1^{o\top} v'/\tau)}$$

is the softmax results over similarities between $u_1^o$ and other samples, and

$$t_v = \frac{\exp(v^\top u_1^o/\tau)}{\sum_{v' \in \mathcal{V}/v} \exp(v^\top v'/\tau)}$$

is computed over similarities between sample $v$ and its contrastive samples $\mathcal{V}/v$. We can see that the first term of $\partial L/\partial u_1^o$ comes from the part which takes $u_1^o$ as the anchor, and the second term comes from the part which takes the other feature as the anchor. Tao et al. (2022) proposes to stop the second term and verifies that stopping the second gradient term will not affect the performance empirically.

Note that the $u_1^t$ term in the gradient is from the alignment loss. So the gradient of the uniformity loss on $u_1^o$ can be written as

$$\frac{\partial L_{\text{uniform}}}{\partial u_1^o} = \frac{1}{2\tau n}\left(\sum_{v \in \mathcal{V}/u_1^o} s_v v\right) + \frac{1}{2\tau n}\left(\sum_{v \in \mathcal{V}/u_1^o} t_v v\right) \tag{11}$$

It is noteworthy that by writing $\mathcal{V} = \{h_1, h_2, \ldots, h_N\}$, Eq(4) shares the same form as Eq(11). By adopting the stop-gradient method just as (Tao et al., 2022) takes, we remove the second term in Eq(4), which is just the $\boldsymbol{A}\boldsymbol{D}^{-1}$ term in Eq(6).

Empirically, we draw the singular value distribution of embeddings for vanilla BERT and +Contra-Norm with only $\boldsymbol{A}\boldsymbol{D}^{-1}$ term or $\boldsymbol{D}^{-1}\boldsymbol{A}$ term on RTE task. As shown in Fig. 4, compared with vanilla BERT with a long-tail distribution (dimensional collapse), adding ContraNorm with $\boldsymbol{D}^{-1}\boldsymbol{A}$ and $\boldsymbol{A}\boldsymbol{D}^{-1}$ both reduce the number of insignificant (nearly zero) singular values and make a more balanced distribution. The similar singular value distributions mean that they play a similar role in alleviating dimensional collapse.

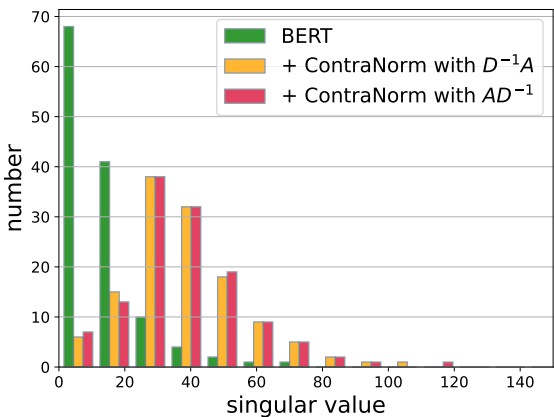

Figure 4: Singular value distributions with ContraNorm using only $\boldsymbol{A}\boldsymbol{D}^{-1}$ term or $\boldsymbol{D}^{-1}\boldsymbol{A}$ term. Experiments are conducted with 12-layer BERT on RTE task.

## B Proofs for Propositions in Section 4.3

We give out two lemmas.

**Lemma B.1.** Denote $\boldsymbol{e} = [1, 1, \ldots, 1]^\top / \sqrt{n}$. For any update form $\boldsymbol{X}_1 = \boldsymbol{P}\boldsymbol{X}_0$ and $\lambda > 0$, if the eigenvalues of $(\boldsymbol{I} - \boldsymbol{e}\boldsymbol{e}^\top) - \lambda \boldsymbol{P}^\top (\boldsymbol{I} - \boldsymbol{e}\boldsymbol{e}^\top)\boldsymbol{P}$ are all not larger than zero, then we have $\mathrm{Var}(\boldsymbol{X}_1) \geq \lambda^{-1} \cdot \mathrm{Var}(\boldsymbol{X}_0)$.

*Proof.* We denote $\boldsymbol{X}_0\boldsymbol{X}_0^\top = \boldsymbol{Y}\mathrm{diag}(\omega_1, \ldots, \omega_n)\boldsymbol{Y}^\top$ for the eigen-decomposition of $\boldsymbol{X}_0\boldsymbol{X}_0^\top$, where $\boldsymbol{Y} = [\boldsymbol{y}_1, \boldsymbol{y}_2, \ldots, \boldsymbol{y}_n]$ is the orthogonal basis and all $\omega_i \geq 0$. Note that $(\boldsymbol{I} - \boldsymbol{e}\boldsymbol{e}^\top)^2 = (\boldsymbol{I} - \boldsymbol{e}\boldsymbol{e}^\top)$. Therefore,

$$
\begin{aligned}
\mathrm{Var}(\boldsymbol{X}_0) &= \|(\boldsymbol{I} - \boldsymbol{e}\boldsymbol{e}^\top)\boldsymbol{X}_0\|_F^2 \\
&= tr\{|(\boldsymbol{I} - \boldsymbol{e}\boldsymbol{e}^\top)\boldsymbol{X}_0\boldsymbol{X}_0^\top(\boldsymbol{I} - \boldsymbol{e}\boldsymbol{e}^\top)\} \\
&= tr\{(\boldsymbol{I} - \boldsymbol{e}\boldsymbol{e}^\top)\boldsymbol{Y}\mathrm{diag}(\omega_1, \ldots, \omega_n)\boldsymbol{Y}^\top(\boldsymbol{I} - \boldsymbol{e}\boldsymbol{e}^\top)\} \\
&= tr\{\mathrm{diag}(\omega_1, \ldots, \omega_n)\boldsymbol{Y}^\top(\boldsymbol{I} - \boldsymbol{e}\boldsymbol{e}^\top)(\boldsymbol{I} - \boldsymbol{e}\boldsymbol{e}^\top)\boldsymbol{Y}\} \\
&= tr\{\mathrm{diag}(\omega_1, \ldots, \omega_n)\boldsymbol{Y}^\top(\boldsymbol{I} - \boldsymbol{e}\boldsymbol{e}^\top)\boldsymbol{Y}\} \\
&= \sum_{i=1}^n \omega_i \boldsymbol{y}_i^\top(\boldsymbol{I} - \boldsymbol{e}\boldsymbol{e}^\top)\boldsymbol{y}_i
\end{aligned} \tag{12}
$$

Similarly, we have

$$
\mathrm{Var}(\boldsymbol{X}_1) = \sum_{i=1}^n \omega_i \boldsymbol{y}_i^\top \boldsymbol{P}(\boldsymbol{I} - \boldsymbol{e}\boldsymbol{e}^\top)\boldsymbol{P}\boldsymbol{y}_i. \tag{13}
$$

Therefore, we have

$$
\mathrm{Var}(\boldsymbol{X}_0) - \lambda \mathrm{Var}(\boldsymbol{X}_1) = \sum_{i=1}^n \omega_i \boldsymbol{y}_i^\top \{(\boldsymbol{I} - \boldsymbol{e}\boldsymbol{e}^\top) - \lambda \boldsymbol{P}^\top(\boldsymbol{I} - \boldsymbol{e}\boldsymbol{e}^\top)\boldsymbol{P}\}\boldsymbol{y}_i. \tag{14}
$$

Thus, if the eigenvalues of $(\boldsymbol{I} - \boldsymbol{e}\boldsymbol{e}^\top) - \lambda \boldsymbol{P}^\top(\boldsymbol{I} - \boldsymbol{e}\boldsymbol{e}^\top)\boldsymbol{P} := \boldsymbol{\Sigma}$ are all not larger than zero, $\boldsymbol{\Sigma}$ is semi-negative definite, then we have

$$
\boldsymbol{y}_i^\top \{(\boldsymbol{I} - \boldsymbol{e}\boldsymbol{e}^\top) - \lambda \boldsymbol{P}^\top(\boldsymbol{I} - \boldsymbol{e}\boldsymbol{e}^\top)\boldsymbol{P}\}\boldsymbol{y}_i \leq 0, \tag{15}
$$

which implies that $\mathrm{Var}(\boldsymbol{X}_0) - \lambda \mathrm{Var}(\boldsymbol{X}_1) \leq 0$. Therefore, $\mathrm{Var}(\boldsymbol{X}_1) \geq \lambda^{-1} \cdot \mathrm{Var}(\boldsymbol{X}_0)$. $\square$

The second lemma is from the Eq.(13) in (Fulton, 2000).

**Lemma B.2.** Let $A, B, C$ be symmetric matrices and $C = A + B$. Suppose the eigenvalues of $A$ are $\alpha_1 \geq \alpha_2 \geq \cdots \geq \alpha_n$, the eigenvalues of $B$ are $\beta_1 \geq \beta_2 \geq \cdots \geq \beta_n$, and the eigenvalues of $C$ are $\gamma_1 \geq \gamma_2 \geq \cdots \geq \gamma_n$. Then we have the inequality

$$\max_{i+j=n+k} \alpha_i + \beta_j \leq \gamma_k \leq \min_{i+j=k+1} \alpha_i + \beta_j. \tag{16}$$

We can now start to prove Proposition 1.

**Proposition 1.** Let $e = (1, 1, \ldots, 1)^\top / \sqrt{n}$. For attention matrix $\bar{A} = \text{softmax}(H_b H_b^\top)$, let $\sigma_{\min}$ be the smallest eigenvalue of $P = (I - ee^\top)(I - \bar{A}) + (I - \bar{A})^\top(I - ee^\top)$. For the ContraNorm update $H_t = ((1 + s)I - s\bar{A})H_b$, $s > 0$, we have $Var(H_t) \geq (1 - s\sigma_{\min})^{-1} \cdot Var(H_b)$. Especially, if $\sigma_{\min} \geq 0$, we have $Var(H_t) \geq Var(H_b)$.

*Proof.* We denote $\Sigma = (I - ee^\top) - \lambda((1+s)I - s\bar{A})^\top(I - ee^\top)((1+s)I - s\bar{A})$. Then,

$$\Sigma = (1-\lambda)I - \lambda(s(I - \bar{A})^\top(I - ee^\top) - s(I - ee^\top)(I - \bar{A}) - s^2(I - \bar{A})^\top(I - ee^\top)(I - \bar{A}))$$

$$= (1-\lambda)I - \lambda sP - \lambda s^2(I - \bar{A})^\top(I - ee^\top)(I - \bar{A}). \tag{17}$$

Let $\alpha_1 \geq \alpha_2 \geq \cdots \geq \alpha_n$ be the eigenvalues of $\Sigma$. Since $I - ee^\top$ has a eigenvalue of 0 and $n - 1$ eigenvalues of 1, $I - ee^\top$ is a semi-definite positive matrix. Thus, $s^2(I - \bar{A})^\top(I - ee^\top)(I - \bar{A})$ is also a semi-definite positive matrix. Notice that the largest eigenvalue of $(1 - \lambda)I - sP$ is $(1 - \lambda) - s\sigma_{\min}$ and the largest eigenvalue of $-s^2(I - \bar{A})^\top(I - ee^\top)(I - \bar{A})$ is 0. Therefore, by Lemma B.2, the largest eigenvalue of $\Sigma$ is less or equal to $(1 - \lambda) - s\sigma_{\min}$. Let $\lambda = 1 - s\sigma_{\min}$, then the largest eigenvalue of $\Sigma$ is less or equal to 0. By Lemma B.1, we have $Var(H_t) \geq (1 - s\sigma_{\min})^{-1} \cdot Var(H_b)$.

Moreover, if $\sigma_{\min} \geq 0$, then $(1 - s\sigma_{\min})^{-1} \geq 1$, leading to $Var(H_t) \geq Var(H_b)$. $\qquad\square$

**Remark.** Now we discuss some sufficient conditions for $\sigma_{\min} \geq 0$. If $P$ is a diagonally dominant matrix, then we will have the result $\sigma_{\min} \geq 0$. Denote $a_{ij} = \bar{A}_{ij}$, $b_{ij} = \sigma_{ij} - a_{ij}$ and $Q = (I - ee^\top)(I - \bar{A})$, where $\sigma_{ij} = 1$ if $i = j$ and $\sigma_{ij} = 0$ if $i \neq j$, then we have

$$Q_{ij} = b_{ij} - \frac{1}{n}\sum_{k=1}^{n}(b_{kj}). \tag{18}$$

If we have $\sum_{k=1}^{n} a_{kj} \leq 1 + na_{ij}$ for any $i, j$, then we will have

$$b_{jj} \geq \frac{\sum_{k=1}^{n} b_{kj}}{n} \geq b_{ij}, \ i \neq j. \tag{19}$$

Notice that $\sum_{k=1}^{n} b_{kj} = 0$, we have

$$|Q_{jj}| = |\sum_{k\neq j} Q_{kj}| \tag{20}$$

Since $\bar{A}$ is an attention matrix, we also have

$$|Q_{jj}| = |\sum_{k\neq j} Q_{jk}|. \tag{21}$$

Therefore, we have

$$\begin{aligned}
|P_{jj}| &= 2|Q_{jj}| \\
&= |\sum_{k\neq j} Q_{kj}| + |\sum_{k\neq j} Q_{jk}| \\
&\geq |\sum_{k\neq j} Q_{kj} + \sum_{k\neq j} Q_{jk}| \\
&\geq |\sum_{k\neq j} P_{kj}|,
\end{aligned} \tag{22}$$

which indicates that $P$ is a diagonally dominated matrix, thus $\sigma_{\min} \geq 0$. Therefore, $\sum_{k=1}^{n} a_{kj} \leq 1 + na_{ij}$ for any $i, j$ is just a sufficient condition. A special case for this is $\sum_{k=1}^{n} a_{kj} = 1, \forall k$.

We now move on to the proof of Proposition 2. We first give a lemma on the property of effective rank.

**Lemma B.3.** Let the eigenvalues of $AA^\top$ be $\lambda_1 \geq \lambda_2 \geq \cdots \geq \lambda_n$ and the eigenvalues of $BB^\top$ be $\sigma_1 \geq \sigma_2 \geq \ldots \sigma_n$. If $\sigma_i / \lambda_i$ is increasing as $i$ increases, then we have $\mathrm{erank}(B) \geq \mathrm{erank}(A)$.

This lemma can be proved just by using the definition of the effective rank.

**Proposition 2.** Consider the update form

$$\boldsymbol{H}_t = (1+s)\boldsymbol{H}_b - s(\boldsymbol{H}_b \boldsymbol{H}_b^\top)\boldsymbol{H}_b, \tag{23}$$

let $\sigma_{\max}$ be the largest singular value of $\boldsymbol{H}_b$. For $s > 0$ satisfying $1 + (1 - \sigma_{\max}^2)s > 0$, we have $\mathrm{erank}(\boldsymbol{H}_t) > \mathrm{erank}(\boldsymbol{H}_b)$.

*Proof.* We have

$$\boldsymbol{H}_t = ((1+s)\boldsymbol{H}_b - s\boldsymbol{H}_b \boldsymbol{H}_b^\top)\boldsymbol{H}_b. \tag{24}$$

Therefore,

$$\boldsymbol{H}_t \boldsymbol{H}_t^\top = s^2(\boldsymbol{H}_b \boldsymbol{H}_b^\top)^3 - 2s(1+s)(\boldsymbol{H}_b \boldsymbol{H}_b^\top)^2 + (1+s)^2(\boldsymbol{H}_b \boldsymbol{H}_b^\top) \tag{25}$$

Suppose $\lambda_1 \geq \lambda_2 \geq \cdots \geq \lambda_n$ are the eigenvalues of $\boldsymbol{H}_b \boldsymbol{H}_b^\top$, then $\boldsymbol{H}_t \boldsymbol{H}_t^T$ has the same eigenvectors as $\boldsymbol{H}_b \boldsymbol{H}_b^\top$, and its eigenvalues are $(\lambda_i s - (1+s))^2 \lambda_i$. Since $s$ satisfies $1 + (1 - \sigma_{\max}^2 s) > 0$, we have $1 + s > \lambda_i s$. Therefore, $(\lambda_i s - (1+s))^2$ is increasing as $i$ increases, resulting the fact that $\mathrm{erank}(\boldsymbol{H}_t) > \mathrm{erank}(\boldsymbol{H}_b)$ by using Lemma B.3. $\square$

## C  METRICS CALCULATING FEATURE AND ATTENTION MAP COSINE SIMILARITY

Following Wang et al. (2022), given feature map $\boldsymbol{H} \in \mathbb{R}^{N \times D}$ and attention map of the $h$-th head $\boldsymbol{A}^{(h)} \in \mathbb{R}^{N \times N}$ with batch size $N$ and hidden embedding size $D$, the feature cosine similarity and the attention cosine similarity is computed as

$$\mathrm{sim}_f = \frac{2}{N(N-1)} \sum_{i,j>i} \frac{\boldsymbol{h}_i^\top \boldsymbol{h}_j}{\|\boldsymbol{h}_i\|_2 \|\boldsymbol{h}_j\|_2}, \quad \mathrm{sim}_{\mathrm{attn}} = \frac{2}{N(N-1)H} \sum_{i,j>i} \frac{\boldsymbol{a}_i^{(h)\top} \boldsymbol{a}_j^{(h)}}{\|\boldsymbol{a}_i^{(h)}\|_2 \|\boldsymbol{a}_j^{(h)}\|_2},$$

where $\boldsymbol{h}_i$ denotes the $i$-th row of $\boldsymbol{H}$, $\boldsymbol{a}_i$ is the $i$-th column of $\boldsymbol{A}$, and $H$ is the number of attention heads.

## D  APPLYING THE UNIFORMITY LOSS DIRECTLY

We conduct a comparative experiment on BERT model with straightforwardly applied uniformity loss and our proposed ContraNorm. Specifically, we add the uniformity loss ($loss_{\mathrm{uni}}$) to the classification loss (MSELoss, CrossEntropyLoss or BCELoss depending on the task type, denoted by $loss_{\mathrm{cls}}$). Formally, the final loss is written as

$$loss_{\mathrm{total}} = loss_{\mathrm{cls}} + loss_{\mathrm{uni}},$$

where $loss_{\mathrm{uni}} = \sum_{i=1}^{N} \log \sum_{j=1}^{N} \exp(f(\mathbf{x}_i)^T f(\mathbf{x}_j)/\tau)$ and $N$ is the number of samples in the batch. We tune $\tau$ in the range of $[0.5, 0.8, 1.0, 1.2, 1.5]$ and choose the best one in terms of average performance. Other hyperparameters are kept the same as settings of ContraNorm. The results are shown in the Table.5

We can see that +ContraNorm gets the best score in 8 / 9 tasks, while +Uniformity loss reaches the best only in 1 / 9 tasks. ContraNorm also has the highest average score among all tasks. The reason is that updating the total loss is a combined process for objectives of correct classification and uniform distribution. Thus, a lower $loss_{\mathrm{total}}$ may be only caused by a lower classification loss while uniformity loss is kept the same, which cannot ensure a more uniform distribution of representations. In contrast, ContraNorm acts directly on representations in each layer and enforces the uniform property.

In fact, there are many methods in GNNs such as Yang et al. (2021) and Zhu et al. (2021), which design the propagation mechanism under the guidance of the corresponding objective. The well-designed propagation mechanism is shown to be the most fundamental part of GNNs (Zhu et al.,

Table 5: Results comparison on validation set of GLUE tasks. Following (Devlin et al., 2018), we report F1 scores for QQP and MRPC, Spearman correlations for STS-B, and accuracy scores for the other tasks. **Avg** denotes the average performance on all the tasks. For each task, the best performance is bolded.

| Dataset | COLA | SST-2 | MRPC | QQP | STS-B | MNLI-m | MNLI-mm | QNLI | RTE | Avg |
|---|---|---|---|---|---|---|---|---|---|---|
| BERT-base | 55.28 | 92.89 | 88.96 | 88.24 | 88.61 | 84.65 | 84.61 | 91.51 | 68.59 | 82.59 |
| BERT + Uniformity Loss | 58.08 | 93.00 | 89.46 | 88.14 | **88.69** | 84.45 | 84.43 | 91.60 | 68.59 | 82.94 |
| BERT + ContraNorm | **58.83** | **93.12** | **89.49** | **88.30** | 88.66 | **84.87** | **84.66** | **91.78** | **70.76** | **83.39** |

Table 6: Results comparison on different plugging positions of ContraNorm. Experiments are evaluated on the validation set of GLUE tasks. **Avg** denotes the average performance on all the tasks. The best result for each task is bolded.

| Dataset | COLA | SST-2 | MRPC | QQP | STS-B | MNLI-m | MNLI-mm | QNLI | RTE | Avg |
|---|---|---|---|---|---|---|---|---|---|---|
| BERT-base | 55.28 | 92.89 | 88.96 | 88.24 | 88.61 | 84.65 | 84.61 | 91.51 | 68.59 | 82.59 |
| before-residual | 59.57 | 93.12 | 89.97 | 88.30 | 88.93 | 84.84 | 84.67 | 91.58 | 68.95 | 83.33 |
| after-residual | 58.83 | 93.12 | 89.49 | 88.30 | 88.66 | 84.87 | 84.66 | 91.78 | 70.76 | 83.39 |

2021). Instead of directly using the loss function, these methods transfer the loss function into a specific propagation method and achieve superior performance, which indicates that changing the network may be more effective than directly adding the objective to the loss function.

## E   CHOICES OF CONTRANORM PLUGGING POSITION

We explore two ways to integrate ContraNorm into BERT and ALBERT. Concretely, consider the update of $\boldsymbol{H}^{(l)}$ in $l$-th block of Transformers

$$\boldsymbol{H}^{(l)} = \text{MultiHeadAttention}(\boldsymbol{H}^{(l)}), \tag{26}$$

$$\boldsymbol{H}^{(l)} = \boldsymbol{H}^{(l)} + \boldsymbol{X}, \tag{27}$$

$$\boldsymbol{H}^{(l)} = \text{LayerNorm}(\boldsymbol{H}^{(l)}), \tag{28}$$

where $\boldsymbol{X} = \boldsymbol{H}_b$ is the input tensor.

We choose positions 1) between Eq.(26) and Eq.(27), named as *before-residual*; 2) between Eq.(27) and Eq.(28), named as *after-residual*. The performance comparison between the two positions on GLUE datasets is shown in Table. (6). It is observed that putting ContraNorm after the residual connection slightly outperforms that before residual. Therefore, we choose the after-residual variant as our basic ContraNorm.

## F   TIME COMPLEXITY ANALYSIS AND A MODIFIED CONTRANORM

The most time-consuming operation of ContraNorm is the matrix multiplication. Given $\mathbf{H} \in \mathbb{R}^{n \times d}$, where $n$ and $d$ denote the number of samples in a batch and feature size respectively, the time complexity of ContraNorm is $\mathcal{O}(n^2 d)$, which is the same order as the self-attention operation in Transformer. Empirically, we report the training time of BERT with or without ContraNorm on GLUE tasks in Table 7. All experiments are conducted on a single NVIDIA GeForce RTX 3090. On average, we raise the performance of BERT on GLUE tasks from 82.59% to 83.54% (see Table 1) with less than 4 minutes overhead. We think the time overhead is acceptable considering the benefits it brings.

Moreover, to enhance scalability we propose a modified version of ContraNorm inspired by Ali et al. (2021). As already alluded to, ContraNorm resolves dimensional collapse by pushing samples away from each other with the uniformity loss as the energy function. A different choice of the underlying function generates a new form of normalization layer. Here, to maintain the same goal of settling dimensional collapse, we choose another contrastive loss named Barlow Twins (Zbontar et al., 2021)

$$L_{BT} = \sum_i (1 - C_{ii})^2 + \lambda \sum_i \sum_{j \neq i} C_{ij}^2 \tag{29}$$

Table 7: Estimated training time of BERT with or without ContraNorm on GLUE tasks. All experiments are conducted on a single NVIDIA GeForce RTX 3090. $s$ is the abbreviation for second. **Avg** denotes the average training time across all the tasks.

| Dataset | COLA | SST-2 | MRPC | QQP | STS-B | MNLI-m (/mm) | QNLI | RTE | **Avg** |
|---------|------|-------|------|-----|-------|--------------|------|-----|---------|
| BERT | $110s$ | $851s$ | $74s$ | $6458s$ | $110s$ | $8186s$ | $2402s$ | $74s$ | $2283s$ |
| +ContraNorm | $125s$ | $983s$ | $80s$ | $7150s$ | $122s$ | $8941s$ | $2594s$ | 80s | $2509s$ |

with $C_{ij} = \sum_b f(\mathbf{x}_{b,i}^A) f(\mathbf{x}_{b,j}^B) / \sqrt{\sum_b (f(\mathbf{x}_{b,i}^A))^2} \sqrt{\sum_b (f(\mathbf{x}_{b,j}^B))^2}$, where $\lambda$ is a trading-off constant, $b$ is the batch index, $i, j$ index the vector dimension, and $\mathbf{x}^A, \mathbf{x}^B$ denotes different augmented views of $\mathbf{x}$. Dislike the sample-wise loss InfoNCE, Barlow Twins decorrelates the dimensions and can be seen as contrastive between the dimensions of the embeddings (Garrido et al., 2022). Despite seeming different, recent works empirically show strong similarities between the two loss functions, and theoretically prove the equivalence between them under limited assumptions (Garrido et al., 2022; Tao et al., 2022; Balestriero & LeCun, 2022; Huang et al., 2021). Therefore, the layer generated by unfolding iterations of Barlow Twins is believed to work equally well as ContraNorm. Specifically, we reorder the matrix multiplication from $\mathbf{HH}^T \in \mathbb{R}^{n \times n}$ to $\mathbf{H}^T \mathbf{H} \in \mathbb{R}^{d \times d}$ to enforce the channels of embeddings different from each other. Then the modified version of ContraNorm becomes

$$\text{ContraNorm-v2}(\mathbf{H}) = \text{LayerNorm}(\mathbf{H} - s/\tau \times \mathbf{H} \times \text{softmax}(\mathbf{H}^T \mathbf{H})). \tag{30}$$

The transposed alternative has a linear complexity in the number of samples, i.e. $\mathcal{O}(nd^2)$. In the case of $n \gg d$, the modified version greatly alleviates the computation burden.

To verify whether it performs equally well as ContraNorm, we conduct experiments on the validation set of GLUE tasks. The learning rate of BERT with ContraNorm-v2 is set to $4e - 5$ and other experiment setups are the same in Section 4.1. As shown from Table 8, for each task the performance of BERT with ContraNorm-v2 surpasses the vanilla BERT, and the average performance is raised from $82.59\%$ to $84.21\%$. The results imply effectiveness of this modified version of ContraNorm, which can also be explained with the relationship between Gram matrix ($\mathbf{G} = \mathbf{HH}^T \in \mathbb{R}^{n \times n}$) and covariance matrix ($\mathbf{C} = \mathbf{H}^T \mathbf{H} \in \mathbb{R}^{d \times d}$) (Ali et al., 2021).

Table 8: Results comparison of BERT with or without ContraNorm-v2 on validation set of GLUE tasks. Following Devlin et al. (2018), we report F1 scores for QQP and MRPC, Spearman correlations for STS-B, and accuracy scores for the other tasks. **Avg** denotes the average performance on all the tasks and bold denotes the best performance.

| Dataset | COLA | SST-2 | MRPC | QQP | STS-B | MNLI-m | MNLI-mm | QNLI | RTE | **Avg** |
|---------|------|-------|------|-----|-------|--------|---------|------|-----|---------|
| BERT | 55.28 | 92.89 | 88.96 | 88.24 | 88.61 | 84.65 | 84.61 | 91.51 | 68.59 | 82.59 |
| + ContraNorm-v2 | **62.08** | **93.69** | **91.78** | **88.36** | **89.59** | **85.24** | **85.19** | **91.95** | **70.04** | **84.21** |

# G   RESULTS ON TEST SET OF GLUE DATASETS

We submit our trained model to GLUE benchmark leaderboard and the resultant feedback of performance is shown in Table (9).

Table 9: Results comparison on test set of GLUE tasks. **Avg** denotes the average performance on all the tasks.

| Dataset | COLA | SST-2 | MRPC | QQP | STS-B | MNLI-m | MNLI-mm | QNLI | RTE | **Avg** |
|---------|------|-------|------|-----|-------|--------|---------|------|-----|---------|
| BERT-base | 53.3 | 92.8 | 86.8 | 71.2 | 82.8 | 84.4 | 83.2 | 90.9 | 66.3 | 79.1 |
| BERT + ContraNorm | 54.5 | 93.0 | 87.9 | 71.4 | 83.0 | 84.5 | 83.4 | 91.0 | 66.9 | 79.5 |

## H  INTRODUCTION OF GRAPH DATASETS

**Citation Network.** Cora, CiteSeer are three popular citation graph datasets. In these graphs, nodes represent papers and edges correspond to the citation relationship between two papers. Nodes are classified according to academic topics.

**Wikipedia Network.** Chameleon and Squirrel are Wikipedia page networks on specific topics, where nodes represent web pages and edges are the mutual links between them. Node features are the bag-of-words representation of informative nouns. The nodes are classified into four categories according to the number of the average monthly traffic of the page.

Statics of datasets are shown in Table. (10).

Table 10: Graph datasets statics.

| Category | Dataset | # Nodes | # Edges | # Features | Degree | # Classes |
|---|---|---|---|---|---|---|
| Citation | Cora | 2708 | 5278 | 1433 | 4.90 | 7 |
| | CiteSeer | 3327 | 4552 | 3703 | 3.77 | 6 |
| Wikipedia | Chameleon | 2277 | 36101 | 500 | 5.0 | 6 |
| | Squirrel | 5201 | 217073 | 2089 | 154.0 | 4 |

## I  ABLATION STUDY FULL RESULTS

We conduct experiments using BERT+ContraNorm with varying scaling factor $s$ on GLUE datasets. For each dataset, we vary the normalization scale around the best choice, and other parameters remain consistent. As shown in Fig.(5), the results illustrate that model with ContraNorm is robust in an appropriate range of normalization scaling factor.

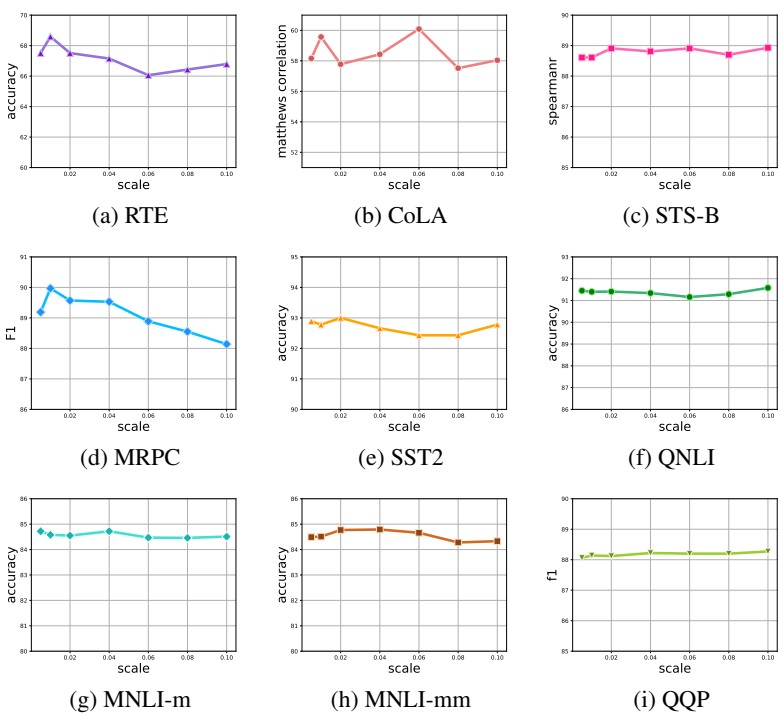

Figure 5: Performance when varying scale factor $s$ on GLUE datasets. We choose BERT as the base model. The varying range is an interval containing the best choice of $s$. Here, we fix it to $[0.005, 0.1]$ for all tasks.

