# OpenReview forum: "ContraNorm: A Contrastive Learning Perspective on Oversmoothing and Beyond"
_ICLR.cc/2023/Conference — ICLR 2023 poster_

### Official Review · Reviewer_uxeV · 2022-10-25

**Confidence:** 4
**Correctness:** 3
**Technical Novelty And Significance:** 2
**Empirical Novelty And Significance:** 2
**Recommendation:** 6

**Clarity, Quality, Novelty And Reproducibility:**

The clarity is clear but I have certain concerns on the claims, the quality is overall good, and the novelty is somewhat incremental. I believe the experiments can be somewhat reproduced based on the descriptions of this paper.

**Strength And Weaknesses:**

**Strengths:**

+ The motivation of the proposed method is clear. It is good this paper try to connect the designed normalization method with the uniformity loss, which provides stronger motivation/foundation for this paper, even though I have concerns for its connections and analyses.

+ The effectiveness of the proposed method over baseline is empirically verified in several architectures (e.g., GNN, Transformer) on several datasets.

**Weaknesses:**

1.One severe problem of this paper is that it misses several important related work/baselines to compare[1,2,3,4], either in discussion [1,2,3,4]or experiments[1,2]. This paper addresses to design a normalization layer that can be plugged in the network for avoiding the dimensional collapse of representation (in intermediate layer). This idea has been done by the batch whitening methods [1,2,3] (e.g, Decorrelated Batch Normalization (DBN), IterNorm, etal.). Batch whitening, which is a general extent of BN that further decorrelating the axes, can ensure the covariance matrix of the normalized output as Identity (IterNorm can obtain an approximate one). These normalization modules can surely satisfy the requirements these paper aims to do. I noted that this paper cites the work of Hua et al, 2021, which uses Decorrelated Batch Normalization for Self-supervised learning (with further revision using shuffling). This paper should note the exist of Decorrelated Batch Normalization. Indeed, the first work to using whitening for self-supervised learning is [4], where it shows how the main motivations of whitening benefits self-supervised learning.

2.I have concerns on the connections and analyses, which is not rigorous for me. Firstly, this paper removes the $AD^{-1}$ in Eqn.6, and claims that “In fact, the operation corresponds to the stop-gradient technique, which is widely used in contrastive learning methods (He et al., 2020; Grill et al., 2020). By throwing away some terms in the gradient, stop-gradient makes the training process asymmetric and thus avoids representation collapse with less computational overhead. It verifies the feasibility of our discarding operation”. I do not understand how to stop gradients used in SSL can be connected to the removement of $AD^{-1}$, I expect this paper can provide the demonstration or further clarification.
Secondly, It is not clear why layerNorm is necessary. Besides, how the layer normalization can be replace with an additional factor (1+s) to rescale H shown in claims “For the convenience of analysis, we replace the layer normalization with an additional factor 1 + s to rescale H”. I think the assumption is too strong.
In summary, the connections between the proposed contraNorm and uniformity loss requires: 1) removing $AD^{-1}$ and 2) add layer normalization, furthermore the propositions for support the connection require the assumption “layer normalization can be replace with an additional factor (1+s) to rescale H”. I personally feel that the connection and analysis are somewhat farfetched.



**Other minors:**

1)Figure 1 is too similar to the Figure 1 of Hua et al, 2021, I feel it is like a copy at my first glance, even though I noted some slightly differences when I carefully compare Figure 1 of this paper to Figure 1 of Hua et al, 2021.

2)The derivation from Eqn. 3 to Eqn. 4 misses the temperature $\tau$, $\tau$ should be shown in a rigorous way or this paper mention it.

3)In page 6. the reference of Eq.(24)?

**References:**

[1] Decorrelated Batch Normalization, CVPR 2018
[2] Iterative Normalization: Beyond Standardization towards Efficient Whitening, CVPR 2019
[3] Whitening and Coloring transform for GANs. ICLR, 2019
[4]Whitening for Self-Supervised Representation Learning, ICML 2021


**Summary Of The Paper:**

This paper addresses the overs-moothing problem of graph neural network (GNN) and Transformers, and proposes to avoid the dimensional collapses in the representation, borrowing the idea from self-supervised learning. It thus proposes a normalization layer ContraNorm, which aims to learn a more uniform distribution in the embedding space that can ensures slighter dimensional collapse, motivated on the uniform loss from self-supervised learning. The proposed normalization layer can be inserted into GNNs and Transformers, and improves these baselines based on the experiments on three different scenarios: ViT for image classification, BERT for natural language understanding, and GNNs for node classifications.

**Summary Of The Review:**

This paper is a well-motivated paper with empirical success in experiments, but it misses several important related work for comparison and some claims are not well clarified. In summary, I am slightly negative to this paper.

==================after rebuta==================

I look over the revised paper, and the authors almost address all my concerns. E.g., add the comparison to the missing related works, and re-organize the clarity. I still have concerns on the claims that “In fact, the operation corresponds to the stop-gradient technique, which is widely used in contrastive learning methods”. I believe removing $AD^{-1}$ may improve the performance based on the authors experiments, but it does not correspond to the stop-gradient in contrastive learning (they are different mechanisms, e.g., stop-gradient is essential to prevent collapse in simsiam/BOYL, while I believe removing $AD^{-1}$ is not the essential part in the proposed method.). Even though this concern holds, the revised version is more rigorous than the previous one, and I tend to positive to this paper. I thus raise my score from 5 to 6.

---

> ### Author Response · Authors · 2022-11-15
> **Response to Reviewer uxeV (3/3)**
>
> **Q3.** It is not clear why the layer normalization is necessary. Besides, it needs to demonstrate why layer normalization can be replaced with an additional factor $(1+s)$ to rescale $\mathbf{H}$.
>
> **A3.** We have reorganized this part (Section 3.3) in our revised version. We denote $\mathbf{H}_b$ for the representation matrix before the update and $\mathbf{H}_t$ for the representation matrix after the update. In the new version, in order to solve the problem that the scale of $\mathbf{H}$ may become smaller after the update, we propose two methods:
> - **a)** The first method is to add a regularization term $-\frac{1}{2}\sum_i\|h_i\|_2^2$ to the uniformity loss. When the regularization term becomes smaller, the norm of $h_i$ becomes larger. Therefore, adding this term can help prevent the norm of representations $h_i$ from becoming smaller. In this way, the update form becomes
> \begin{equation}
>     \mathbf{H}_t = (1 + s) \mathbf{H}_b - s/\tau \times \text{softmax}(\mathbf{H}_b\mathbf{H}_b^{\top})\mathbf{H}_b.
> \end{equation}
> We then conduct the theoretical analysis on this update form, showing the fact that it will make the representations become more uniform.
>
> - **b)** The second method draws support from layer normalization, which adjusts the representation according to its mean and variance The update form of the layer normalization on representation $h_i$ is
> $$\text{LayerNorm}(h_i)=\gamma\cdot((h_i-\text{mean}(h_i))/\sqrt{\text{Var}(h_i)+\varepsilon})+\beta,$$ where $\gamma$ and $\beta$ are learnable parameters and $\varepsilon=10^{-5}$ prevents the denominator from becoming zero. The learnable parameters $\gamma$ and $\beta$ can rescale the representation $h_i$ to help ease the problem. We append the layer normalization:
> \begin{equation}
>     \mathbf{H}_t = \mbox{LayerNorm}\left(\mathbf{H}_b - s/\tau \times \text{softmax}(\mathbf{H}_b\mathbf{H}_b^{\top})\mathbf{H}_b\right).
> \end{equation}
>
> We empirically compare the two proposed methods and find their performance comparable, while the second one performs slightly better, which is shown in Table 4 in the paper. Therefore, we adopt the second update form and name it Contrastive Normalization (**ContraNorm**).
>
> ---
>
> **Q4.** There are some concerns about Figure 1.
>
> **A4.** We have replaced the original Figure 1 with an illustration of how our proposed ContraNorm solves the dimensional collapse.
>
> ---
>
> **Q5.** The derivation from Eq.3 to Eq.4 misses the temperature $\tau$.
>
> **A5.** Thanks for pointing it out. We have added the temperature $\tau$ in the follow-up equations in the updated version.
>
> ---
>
> **Q6.** In page 6, the reference of Eq.24 seems to have some problems.
>
> **A6.** Thanks for pointing it out. It should be Eq.10. We have fixed this typo in the updated version.
>
>
> ---
>
> Thanks for your comments and hope our answers could address your concerns. We have also revised our paper accordingly. Please let us know if you have additional questions.

---

> > ### Comment · Reviewer_uxeV · 2022-12-10
> > **Comment after the rebuttal**
> >
> > I thank for the authors' response. I look over the revised paper, and the authors almost address all my concerns. E.g., add the comparison to the missing related works, and re-organize the clarity. I still have concerns on the claims that “In fact, the operation corresponds to the stop-gradient technique, which is widely used in contrastive learning methods”. I believe removing $AD^{-1}$ may improve the performance based on the authors experiments, but it does not correspond to the stop-gradient in contrastive learning (they are different mechanisms, e.g., stop-gradient is essential to prevent collapse in simsiam/BOYL, while I believe removing $AD^{-1}$ is not the essential part in the proposed method.). Even though this concern holds, the revised version is more rigorous than the previous one, and I tend to positive to this paper. I thus raise my score from 5 to 6.

---

> ### Author Response · Authors · 2022-11-15
> **Response to Reviewer uxeV (2/3)**
>
> **Q2.** I do not understand how the stop gradients used in SSL can be connected to the removement of $\mathbf{A}\mathbf{D}^{-1}$.
>
> **A2.** We take SimCLR[5] as the contrastive learning framework. Tao et al.[6] has studied the stop-gradient form of SimCLR and illustrates that the stop-gradient operation will make a similar performance to the original one. Based on this, we will elaborate on how $\mathbf{A}\mathbf{D}^{-1}$ is removed in the following point. In fact, we can directly illustrate how the second term in Eq.(4) can be omitted.
> $$    \frac{\partial \hat{L}_{\rm{uniform}}}{\partial h_i} = \sum_j \frac{\exp(h_i^{\top} h_j/\tau)}{\sum_k \exp(h_i^{\top} h_k/\tau)} h_j/\tau + \sum_j \frac{\exp(h_i ^{\top}h_j/\tau)}{\sum_k \exp(h_j ^{\top}h_k/\tau)} h_j/\tau.
> $$
>
> In [6], by denoting the normalized features from the online branch as $u^o_i, i=1,2,\dots,n$, the normalized features from the target branch (although the two branches have no differences in SimCLR) as $u^t_i,i=1,2,\dots,n$ and $\mathcal{V}=\{u_1^o,u_2^o,\dots,u_n^o,u_1^t,u_2^t,\dots,u_n^t\},$ the SimCLR loss can be represented as
> $$L=-\frac{1}{2n}\sum_{u_1\in\mathcal{V}}\log\frac{\exp(u_1^{\top}u_1'/\tau)}{\sum_{u\in\mathcal{V}/u_1}\exp(u_1^{\top}u/\tau)}$$
> where $u_1$ and $u_1'$ are positive pairs and $\tau$ is the temperature. Then the gradient of $L$ on $u_1^o$ can be calculated as
> $$\frac{\partial L}{\partial u_1^o}=\frac{1}{2\tau n}\left(-u_1^t+\sum_{v\in{\mathcal{V}/u_1^o}}s_vv\right)+\frac{1}{2\tau n}\left(-u_1^t+\sum_{v\in\mathcal{V}/u_1^o}t_vv\right)$$
> where
> $$s_v=\frac{\exp(u_1^{o\top}v/\tau)}{\sum_{v'\in\mathcal{V}/u_1^o}\exp(u_1^{o\top}v'/\tau)}$$
> is the softmax results over similarities between $u_1^o$ and other samples, and
> $$t_v=\frac{\exp(v^{\top}u_1^o/\tau)}{\sum_{v'\in\mathcal{V}/v}\exp(v^{\top}v'/\tau)}$$
> is computed over similarities between sample $v$ and its contrastive samples $\mathcal{V}/v$. We can see that the first term of $\partial L/\partial u_1^o$ comes from the part which takes $u_1^o$ as the anchor, and the second term comes from the part which takes the other feature as the anchor. [6] proposes to stop the second term and verifies that stopping the second gradient term will not affect the performance empirically.
>
> Note that the $u_1^t$ term in the gradient is from the alignment loss. So the gradient of the uniformity loss on $u_1^o$ can be written as
> $$\frac{\partial L_{\rm{uniform}}}{\partial u_1^o}=\frac{1}{2\tau n}\left(\sum_{v\in{\mathcal{V}/u_1^o}}s_vv\right)+\frac{1}{2\tau n}\left(\sum_{v\in\mathcal{V}/u_1^o}t_vv\right)$$
> It is noteworthy that by writing $\mathcal{V}=\{h_1,h_2,\dots,h_N\}$, Eq(4) shares the same form as the above equation. By adopting the stop-gradient method just as [6] takes, we remove the second term in Eq(4), which is just the $\mathbf{A}\mathbf{D}^{-1}$ term in Eq(6).
>
> Empirically, we draw the singular value distribution of embeddings for vanilla BERT and +ContraNorm with only $\mathbf{A}\mathbf{D}^{-1}$ term or $\mathbf{D}^{-1}\mathbf{A}$ term. The figure is attached to Appendix A in the revised paper and can be also viewed through [https://ibb.co/BGQjCSw](https://ibb.co/BGQjCSw). As shown in the figure, compared with vanilla BERT with a long-tail distribution (dimensional collapse), adding ContraNorm with $\mathbf{A}\mathbf{D}^{-1}$ and $\mathbf{D}^{-1}\mathbf{A}$ both reduce the number of insignificant (nearly zero) singular values and make a more balanced distribution. The similar singular value distributions mean they play a similar role in alleviating dimensional collapse.
>
> **The explanation on the stop-gradient operation can be found in Appendix A of our revised paper.**
>
> [5] A Simple Framework for Contrastive Learning of Visual Representations, ICML 2020
>
> [6] Exploring the Equivalence of Siamese Self-Supervised Learning via A Unified Gradient Framework, CVPR 2022
>
> ---

---

> ### Author Response · Authors · 2022-11-15
> **Response to Reviewer uxeV (1/3)**
>
> We thank Reviewer uxeV for careful reading and constructive comments. We have fixed the typos that you mention and we will address your concerns in the following points.
>
> ---
>
> **Q1.** Several important related works are missed.
>
> **A1.** Thanks for your suggestion. Whitening methods ensure the covariance matrix of normalized outputs is a diagonal matrix and makes dimensions mutually independent of each other, which implicitly solves dimensional collapse. [1], [2], [3] and [4] all adopt the idea of whitening, but differ in calculating details of the whitening matrix and application domain. Compared with them, we avoid complicated calculations on the squared root of the inverse covariance matrix and the delicate design of backward passes for differentiability. Moreover, [1] and [2] are proposed for convolution operations, [3] is for GAN, and [4] works in self-supervised learning. In contrast, we borrow the idea from contrastive learning and solve oversmoothing in completely different fields like Transformers and GNNs. **We have added this part to the related work in the revision.**
>
> We additionally conduct experiments using DBN. Here we adapt DBN to fit the structure of ViT and test on CIFAR10. As shown in the following table, our ContraNorm shows better performance than DBN.
>
> |Vanilla ViT | +DBN | +ContraNorm |
> |:----------:|:----:|:-----------:|
> |84.45|82.98|85.67|
>
> Note that DBN is delicately designed for convolution operations based on feature maps, operating on a four-dimensional $\mathbf{X} \in \mathbb{R}^{h\times w \times d \times m}$, where $h$ and $w$ indicate the height and width of the feature maps, and $d$ and $m$ are the number of feature maps and examples. However, Transformers is free of convolution operations and acts directly on inputs such as image patches and sentence samples. Therefore, transferring DBN to Transformers is not a trivial thing and thus somewhat explains why the performance here is relatively unsatisfying. Moreover, DBN relies on singular value decomposition(SVD), which suffers from low computing efficiency. DBN also uses the running average to calculate the population mean and whitening matrix, thus needing redundant memory cost. In contrast, ContraNorm avoids complicated SVD operation and extra memory storage.
>
> [1] Decorrelated Batch Normalization, CVPR 2018
>
> [2] Iterative Normalization: Beyond Standardization towards Efficient Whitening, CVPR 2019
>
> [3] Whitening and Coloring transform for GANs. ICLR, 2019
>
> [4] Whitening for Self-Supervised Representation Learning, ICML 2021
>
> ---

---

### Official Review · Reviewer_Q9GP · 2022-10-26

**Confidence:** 5
**Correctness:** 4
**Technical Novelty And Significance:** 2
**Empirical Novelty And Significance:** 2
**Recommendation:** 6

**Clarity, Quality, Novelty And Reproducibility:**

the paper is well written, and very clean.


**Strength And Weaknesses:**

Strength:
It’s a nice method with good motivations and theoretical insights.
The paper is generally well written.
The experiments show the efficacy of the model. It indeed can improve the performance in general.

weakness

1, I would ask one particular extreme case on any dataset: the authours can train a very deeper model that encounter the collapse issue, and the model performance is not good. However, with the help of ContraNorm layer, the issue can be solved. This may be important to show the efficacy.

2, Eq.(2) uniform loss part:  what’s the relation between symbol i^+ and x_k? seems some typos here.

3, could the authours give more reasons/motivations why directly discarding the second terms in Eq. (6)? And “Empirically, the two terms play a similar role in our method.” for any empirical evidence?

4, After reading the paepr, I feel that the ContraNorm is not necessarilty related to/address the collapse issues. As least, there is not enough evidence now. Some visualization of Fig. 1 in real world dataset and backbone is thus needed to directly show that the collapse issues happened when training the model; and ContraNorm can change the distribution of features. This can be shown on any dataset. Otherwise, we can just take ContraNorm as one typical type of contrast learning strategy to better train the model.
 And particularly, note that ContraNorm is built upon the layer norm.


**Summary Of The Paper:**

This paper studies the phenomenon of oversmoothing in GNNs and transformers; and presents the ContraNorm layer. The experiments on real-world dataset show the efficacy of this method.

**Summary Of The Review:**

please refer to the weakness part. I would ask for some direct evidence and visualization that ContraNorm can solve the collapse issue.

---

> ### Author Response · Authors · 2022-11-15
> **Response to Reviewer Q9GP (3/3)**
>
> **Q4.** Is there any evidence or visualization for that ContraNorm can solve the collapse issues?
>
> **A4.** We draw the singular value distributions of embeddings in the last block of 12-layer BERT with or without ContraNorm on the RTE task, which can be viewed in **Fig.2c** of our revised paper or through [https://ibb.co/tXYt839](https://ibb.co/tXYt839).
>
> As shown in this figure, for vanilla BERT, the insignificant (nearly zero) values dominate the singular value distribution in the deep layer, which means representations reside on a low-dimensional manifold and dimensional collapse happens. After adding ContraNorm, the long-tail phenomenon is greatly relieved with larger singular values. The singular value distribution shift shows the effectiveness of ContraNorm in drawing representations to a higher dimensional space and alleviating dimensional collapse.
>
> ---
>
> Thanks for your comments and hope our answers could address your concerns. We have also revised our paper accordingly. Please let us know if there is more to clarify.

---

> ### Author Response · Authors · 2022-11-15
> **Response to Reviewer Q9GP (2/3)**
>
> **Q3.** Could the authors give more reasons/motivations why directly discarding the second term in Eq.(6)? Is there any empirical evidence that "The two terms play a similar role in our method"?
>
> **A3.** We take SimCLR[1] as the contrastive learning framework. Tao et al.[2] has studied the stop-gradient form of SimCLR and illustrates that the stop-gradient operation will make a similar performance to the original one. Based on this, we will elaborate on how $\mathbf{A}\mathbf{D}^{-1}$ is removed in the following point. In fact, we can directly illustrate how the second term in Eq.(4) can be omitted.
> $$    \frac{\partial \hat{L}_{\rm{uniform}}}{\partial h_i} = \sum_j \frac{\exp(h_i^{\top} h_j/\tau)}{\sum_k \exp(h_i^{\top} h_k/\tau)} h_j/\tau + \sum_j \frac{\exp(h_i ^{\top}h_j/\tau)}{\sum_k \exp(h_j ^{\top}h_k/\tau)} h_j/\tau.
> $$
>
> In [2], by denoting the normalized features from the online branch as $u^o_i, i=1,2,\dots,n$, the normalized features from the target branch (although the two branches have no differences in SimCLR) as $u^t_i,i=1,2,\dots,n$ and $\mathcal{V}=\{u_1^o,u_2^o,\dots,u_n^o,u_1^t,u_2^t,\dots,u_n^t\},$ the SimCLR loss can be represented as
> $$L=-\frac{1}{2n}\sum_{u_1\in\mathcal{V}}\log\frac{\exp(u_1^{\top}u_1'/\tau)}{\sum_{u\in\mathcal{V}/u_1}\exp(u_1^{\top}u/\tau)}$$
> where $u_1$ and $u_1'$ are positive pairs and $\tau$ is the temperature. Then the gradient of $L$ on $u_1^o$ can be calculated as
> $$\frac{\partial L}{\partial u_1^o}=\frac{1}{2\tau n}\left(-u_1^t+\sum_{v\in{\mathcal{V}/u_1^o}}s_vv\right)+\frac{1}{2\tau n}\left(-u_1^t+\sum_{v\in\mathcal{V}/u_1^o}t_vv\right)$$
> where
> $$s_v=\frac{\exp(u_1^{o\top}v/\tau)}{\sum_{v'\in\mathcal{V}/u_1^o}\exp(u_1^{o\top}v'/\tau)}$$
> is the softmax results over similarities between $u_1^o$ and other samples, and
> $$t_v=\frac{\exp(v^{\top}u_1^o/\tau)}{\sum_{v'\in\mathcal{V}/v}\exp(v^{\top}v'/\tau)}$$
> is computed over similarities between sample $v$ and its contrastive samples $\mathcal{V}/v$. We can see that the first term of $\partial L/\partial u_1^o$ comes from the part which takes $u_1^o$ as the anchor, and the second term comes from the part which takes the other feature as the anchor. [2] proposes to stop the second term and verifies that stopping the second gradient term will not affect the performance empirically.
>
> Note that the $u_1^t$ term in the gradient is from the alignment loss. So the gradient of the uniformity loss on $u_1^o$ can be written as
> $$\frac{\partial L_{\rm{uniform}}}{\partial u_1^o}=\frac{1}{2\tau n}\left(\sum_{v\in{\mathcal{V}/u_1^o}}s_vv\right)+\frac{1}{2\tau n}\left(\sum_{v\in\mathcal{V}/u_1^o}t_vv\right)$$
> It is noteworthy that by writing $\mathcal{V}=\{h_1,h_2,\dots,h_N\}$, Eq(4) shares the same form as the above equation. By adopting the stop-gradient method just as [2] takes, we remove the second term in Eq(4), which is just the $\mathbf{A}\mathbf{D}^{-1}$ term in Eq(6).
>
> Empirically, we draw the singular value distribution of embeddings for vanilla BERT and +ContraNorm with only $\mathbf{A}\mathbf{D}^{-1}$ term or $\mathbf{D}^{-1}\mathbf{A}$ term. The results are in Appendix A in the revised paper and can be also viewed through [https://ibb.co/BGQjCSw](https://ibb.co/BGQjCSw). As shown in the figure, compared with vanilla BERT with a long-tail distribution (dimensional collapse), adding ContraNorm with $\mathbf{D}^{-1}\mathbf{A}$ and $\mathbf{A}\mathbf{D}^{-1}$ both reduce the number of insignificant (nearly zero) singular values and make a more balanced distribution. The similar singular value distributions mean they play a similar role in alleviating dimensional collapse.
>
> **The explanation on the stop-gradient operation can be found in Appendix A of our revised paper.**
>
> [1] A Simple Framework for Contrastive Learning of Visual Representations, ICML 2020
>
> [2] Exploring the Equivalence of Siamese Self-Supervised Learning via A Unified Gradient Framework, CVPR 2022

---

> ### Author Response · Authors · 2022-11-15
> **Response to Reviewer Q9GP (1/3)**
>
> We thank Reviewer Q9GP for your detailed reading and encouraging comments on our work on its motivation and theoretical insights. Below, we summarize and address your main concerns.
>
> ---
>
> **Q1.** It may be important for showing the efficacy to train a very deeper model with the help of the ContraNorm layer and get good performance, which encounters the collapse issue without using the ContraNorm layer.
>
> **A1.** We show the performance comparison in Figure(3a). The figure shows the performance of different-depth BERTs with or without ContraNorm on the COLA task. When the block number exceeds 24, the performance of BERT w/o ContraNorm degenerates dramatically and drops to nearly zero with a block number higher than 48. However, the application of ContraNorm effectively prevents such deterioration, keeping high performance until blocks as deep as 84. As for the corresponding collapse illustration, we show it in Figure(3c). As can be seen, the effective rank plummets in deeper BERTs without ContraNorm, which indicates a severe dimensional collapse. In contrast, ContraNorm enforces a more uniform sample distribution and prevents such collapse.
>
> ---
>
> **Q2.** There seem to be some typos about the symbol in Eq.(2) uniformity loss part.
>
> **A2.** Thanks for pointing it out. We have revised it in the updated version to $L_{\rm{uniform}}(i)=\log \sum_{k=1}^N\exp(f(\mathbf{x}_i)^{\top}f(\mathbf{x}_k) / \tau).$
>
> ---

---

### Official Review · Reviewer_Zh7A · 2022-10-30

**Confidence:** 3
**Clarity, Quality, Novelty And Reproducibility:** Very clear, good quality. seems repro…
**Correctness:** 3
**Technical Novelty And Significance:** 3
**Empirical Novelty And Significance:** 3
**Recommendation:** 6

**Strength And Weaknesses:**

Comments and questions:
- The paper is very well written with good organization. Its topic of discussion is a well-motivated and focused problem with clear solution which makes it easy to follow. Results, especially for GCN and vision tasks, show good improvement as the depth of models increase.
- Questions:
    - In Subsection 3.1 and Figure 2, it was argued that similarity in attention does not imply feature collapse. The argument is based on using attention map, which is a cosine similarity between query and key, as a measure of representation collapse. The question is, what is referred as feature here? I am assuming the value embedding. If so, similarity between attention map naturally does not imply similarity between value pairs. What is new here or am I missing something?
    - The uniformity term in contrastive learning is between the representation in question and all its negative examples. However, $\dot{L}_{uniform}$ seem to measure similarity across all representation. Is that the case?
     - The normalization layer Eq(7) and Eq(8), are designed as a single gradient step that updates the features in the direction of minimal similarity. Could this be amortized? as opposed to approximating a solution with single step.
- Questions regarding experiment:
     -  What average performance in Table 1?
     -  The average performance for both BERT and ALBERT is reported inconsistently in the **Results** section and **Table 1**. Cloud you correct/clarify?
     -  Reported results in **Table 1** show tiny improvement. Are these the right models and experimental setup to show case your solution for the GLUE task?

- Corrections:
    - missing fullstop in the second line form the last, in the second paragraph of the introduction section.

**Summary Of The Paper:**

The paper discusses an over smoothing issue of representation in Graph Learning and message-passing like computation unit (attention) as the depth of the models increase. The paper argues that attention maps are not a very good indictors of over smoothing, due to the fact that a feature can have low similarity while having high attendance score. Subsequently, an optimization based normalization unit is proposed to encourage large variance across features and decorrelation of dimensions. Results, especially for image and GCN, show good improvement.


**Summary Of The Review:**

The paper motivated the problem attempts to solve, suggested a solution and provided insight into why it would work with theoretical justification. It further showed, empirically, performance improvement as the depth of the model increases.

---

> ### Author Response · Authors · 2022-11-15
> **Response to Reviewer Zh7A (2/2)**
>
> **Q4.** What does the average performance indicate in Table 1?
>
> **A4.** The average performance in Table 1 takes the average of the results on all the nine tasks. We have added the meaning of the average performance in the caption of Table 1.
>
> ---
>
> **Q5.** The average performance for BERT and ALBERT is reported inconsistently.
>
> **A5.** Thanks for pointing it out. The results in the table are the correct ones, while the inline numbers are typos. We have fixed the typos in our revised version.
>
> ---
>
> **Q6.** Are theses the right models and experimental setup to show case your solution for the GLUE task?
>
> **A6.** We follow the standard protocol for evaluating pretrained encoders on GLUE. Specifically, we use the [transformers](https://github.com/huggingface/transformers) library developed by Huggingface, and adopt all default hyperparameters of vanilla BERT and only tune the scaling factor for our ContraNorm. Therefore, it is possible to get better performance after delicately tuning hyper parameters. For example, by simply tuning the learning rate, we raise the performance of STS-B to 89.77% (88.61% for vanilla BERT, 88.66% before tuning), and SST-2 to 93.69% (92.89% for vanilla BERT, 93.12% before tuning).
>
> ---
>
> Thanks for your comments and hope our answers could address your concerns. We have also revised our paper accordingly. Please let us know if you have additional questions.

---

> ### Author Response · Authors · 2022-11-15
> **Response to Reviewer Zh7A (1/2)**
>
> We thank Reviewer Zh7A for careful reading and constructive comments. We have fixed the typos and we address your concerns in the following.
>
> ---
>
> **Q1.** It was argued that the similarity in attention does not imply feature collapse. Similarity between attention map **naturally** does not imply similarity between value pairs. What is new here or am I missing something?
>
> **A1.** The multi-head self-attention modulde in each block is
> \begin{align}
> \mathbf{A}_i &= \text{softmax}(\frac{\mathbf{Q}_i\mathbf{K}_i}{\sqrt{d_k}})\mathbf{V}_i, \\\\
> \mathbf{H} &= \text{Concat}\\{\mathbf{A}_i, \cdots, \mathbf{A}_h\\}\mathbf{W}^O,
> \end{align}
>
> where $\mathbf{Q}_i$, $\mathbf{K}_i$ are query vector, key vector of dimension $d_k$, $\mathbf{V}_i$ is value vector of dimension $d_v$ in the $i$-th attention head, and $\mathbf{W}^O$ is a trainable parameter.
>
>
> The feature here refers to feature embeddings $\mathbf{H}$. The core message to convey in *Figure2(a)* and *2(b)* is that cosine similarity is not a reliable metric to indicate oversmoothing. When talking about oversmoothing of Transformers, prevailing researches focus on a high similarity of **attention matrix**, namely $\mathbf{A}$, which will *intuitively* (as they suppose) cause **embeddings** $\mathbf{H}$ to lose diversity considering the self-attention mechanism. However, in our experiments, we find that cosine similarity of $\mathbf{H}$ is still relatively low. Therefore, in terms of cosine similarity on $\mathbf{H}$, the representations *do not* degenerate, and thus cannot explain the performance deterioration. In contrast, effective rank (namely, the entropy of normalized singular values) proposed in our work depicts the level of representation dimensional collapse and gives a more accurate metric of oversmoothing.
>
>
> ---
>
> **Q2.** The uniformity term in contrastive learning is between the representation in question and all its negative examples. However, $\dot{L}_{\rm{uniform}}$ seems to measure similarity across all representations. Is that the case?
>
> **A2.** We note that in contrastive learning, the total uniformity loss is also computed by summarizing all per-sample uniformity loss (Eq. 2). Therefore, similarly, in our reformulization on graphs, we also first compute per-sample uniformity loss for each node i, and then summarize over all nodes to obtain a total uniformity loss on the graph.
>
>
> ---
>
> **Q3.** The normalization layer Eq(7) and Eq(8) are designed as a single gradient step that updates the features in the direction of minimal similarity. Could this be amortized?
>
> **A3.** We additionaly conduct multi-step updates for minimizing uniformity loss and compare it with single-step method (our ContraNorm). We choose step = 3, 5, 10, and tune the scaling factor $s$ in the range of [0.001, 0.005, 0.01, 0.02, 0.05, 0.1, 0.2] for step-3, step-5, [0.0001, 0.0002, 0.0005, 0.001, 0.005, 0.01, 0.02] for step-10. The results are as follows.
>
> | ContraNorm #Steps   | RTE | COLA | MRPC| STSB | SST2 |QNLI|MNLI-m|MNLI-mm|QQP|Avg| #tasks with highest score |
> |:--:|:--:|:--:|:--:|:--:|:--:|:--:|:--:|:--:|:--:|:--:|:--:|
> | no step (vanilla BERT)| 68.59 | 55.28 | 88.96 | 88.61 | 92.89 | 91.51 | 84.65 | 84.61 | 88.24 | 82.59 | 0 |
> | 1-step (default)| **70.76** | 58.83 | 89.49 | 88.66 | 93.12 | **91.78** | 84.87 | 84.66 | **88.3** | **83.39** | 3 |
> |3-step| 70.04 | **59.62** | 89.23 | **88.96** | 93.12 | 91.56 | 84.84 | 84.62 | 88.28 | 83.36 | 2 |
> |5-step| 68.59 | 59.05 | 89.30 | 88.91 | **93.23** | 91.62 | 84.58 | 84.68 | **88.3** | 83.18 | 2 |
> |10-step| 68.95 | 58.38 | **89.60** | **88.96** | **93.23** | 91.56 | **84.89** | **84.75** | 88.27 | 83.18 | 5 |
>
> In the table, the last column represents how many tasks each method reaches the highest score. It shows that multi-step updates slightly improve the performance on some tasks, e.g., MRPC and MNLI-m. Howerver, in terms of average performance, single-step method is still the best. Considering the multi step updates also bring additional computation time cost, we finally choose to use the single-step update.
>
> ---

---

### Official Review · Reviewer_HpT5 · 2022-10-30

**Confidence:** 4
**Correctness:** 3
**Technical Novelty And Significance:** 3
**Empirical Novelty And Significance:** 2
**Recommendation:** 8

**Clarity, Quality, Novelty And Reproducibility:**

Clarity:

This paper is hard to understand, and the main reason is the notations used in Section 3. I am heavily confused by the $\mathbf{H}$ in Eq. 6~12. It seems that the authors are updating the $\mathbf{H}$, however, I wonder if Eq. 10 still holds for $\mathbf{H}^2$ and $\mathbf{H}^1$, what does the superscript stand for? layers or what else?

Quality:

The overall quality of this paper is not satisfying because of the weaknesses mentioned above.

Novelty:

The idea to convert the contrastive uniformity loss to the forward pass is very sophisticated, the method should be helpful to this community.

Reproducibility:

The paper lacks details on how the methods are actually implemented in the Transformers. For instance, how the stop-gradient is used? How the layer norm is configured in terms of the bias and scale factor? How is the former layer norm dealt with when you apply the contranorm before it?

**Strength And Weaknesses:**

Strength:

The paper presents a nice idea to convert the gradient to the form of normalization.


Weaknesses:

Instinctively, I would like to ask why it is not possible to apply uniformity loss straightforwardly unless it doesn't converge, the authors should provide a comparison between the proposed method and uniformity loss. Since, theoretically, the ContraNorm is equivalent to applying the loss, and appending the ContraNorm to the neural network does increase the computational costs during the inference time.

The authors did not provide an analysis of the computational costs of the proposed Contranorm, I wonder if the number of tokens grows, it will significantly decrease the training speed.

In the ImageNet1K experiment, the performance for #L=24 is still lower than that of #L=16, which means the proposed method does not actually address the mentioned over-smoothing problem, despite the effective rank depicted in figures 2 & 3 seems to be much higher than the reference model. Therefore, I do not consider the mechanism of feature collapsing discussed in this paper to be comprehensive or even correct, which challenges the motivation of this study.

I do not understand how the layer norm can be replaced by the factor 1+s in Eq.9, the layer norm should remove the bias which the factor 1+s cannot.

The clarity of this paper is not acceptable to this conference.

**Summary Of The Paper:**

This paper tackles the collapse problem within nodes (tokens) of a sample if I correctly got the idea of it. The authors employ the effective rank to measure the quailty of the feature. Then, they borrow the idea of uniformity loss from the contrastive learning algorithm, and convert the gradient through back-propagation to the forward-propagation as the Contranorm algorithm. Finally in the experiments, the authors provide several results that the Contranorm boost the performance.

**Summary Of The Review:**

I did really spend a lot of time understanding this paper, and it is not because the technical details are difficult, it is simply because the paper is low in clarity and representation quality. I suggest the authors carefully revise this paper and submit it to another conference.

---

> ### Author Response · Authors · 2022-11-15
> **Response to Reviewer HpT5 (4/4)**
>
> (continuing A6)
>
> As for the stop-gradient, it serves in our theoretical derivation rather than acting as a technical component. Specifically, the removement of $\mathbf{A}\mathbf{D}^{-1}$ in $\mathbf{H}_t = \mathbf{H}_b - s / \tau \times (\mathbf{D}^{-1}\mathbf{A} + \mathbf{A}\mathbf{D}^{-1})\mathbf{H}_b$ (Eq.(6) in the paper) corresponds to stop gradient.
>
> We take SimCLR [3] as the contrastive learning framework. Tao et al. [4] have studied the stop-gradient form of SimCLR and illustrated that the stop-gradient operation will make a similar performance with the original one. Based on this, we will elaborate on how $\mathbf{A}\mathbf{D}^{-1}$ is removed in the following point. In fact, we can directly illustrate how the second term in Eq.(4) can be omitted.
> $$    \frac{\partial \hat{L}_{\rm{uniform}}}{\partial h_i} = \sum_j \frac{\exp(h_i^{\top} h_j/\tau)}{\sum_k \exp(h_i^{\top} h_k/\tau)} h_j/\tau + \sum_j \frac{\exp(h_i ^{\top}h_j/\tau)}{\sum_k \exp(h_j ^{\top}h_k/\tau)} h_j/\tau.
> $$
>
> In [4], by denoting the normalized features from the online branch as $u^o_i, i=1,2,\dots,n$, the normalized features from the target branch (although the two branches have no differences in SimCLR) is $u^t_i,i=1,2,\dots,n$ and $\mathcal{V}=\{u_1^o,u_2^o,\dots,u_n^o,u_1^t,u_2^t,\dots,u_n^t\},$ the SimCLR loss can be represented as
> $$L=-\frac{1}{2n}\sum_{u_1\in\mathcal{V}}\log\frac{\exp(u_1^{\top}u_1'/\tau)}{\sum_{u\in\mathcal{V}/u_1}\exp(u_1^{\top}u/\tau)}$$
> where $u_1$ and $u_1'$ are positive pairs and $\tau$ is the temperature. Then the gradient of $L$ on $u_1^o$ can be calculated as
> $$\frac{\partial L}{\partial u_1^o}=\frac{1}{2\tau n}\left(-u_1^t+\sum_{v\in{\mathcal{V}/u_1^o}}s_vv\right)+\frac{1}{2\tau n}\left(-u_1^t+\sum_{v\in\mathcal{V}/u_1^o}t_vv\right)$$
> where
> $$s_v=\frac{\exp(u_1^{o\top}v/\tau)}{\sum_{v'\in\mathcal{V}/u_1^o}\exp(u_1^{o\top}v'/\tau)}$$
> is the softmax results over similarities between $u_1^o$ and other samples, and
> $$t_v=\frac{\exp(v^{\top}u_1^o/\tau)}{\sum_{v'\in\mathcal{V}/v}\exp(v^{\top}v'/\tau)}$$
> is computed over similarities between sample $v$ and its contrastive samples $\mathcal{V}/v$. We can see that the first term of $\partial L/\partial u_1^o$ comes from the part which takes $u_1^o$ as the anchor, and the second term comes from the part which takes the other feature as the anchor. [4] proposes to stop the second term and verifies that stopping the second gradient term will not affect the performance empirically.
>
> Note that the $u_1^t$ term in the gradient is from the alignment loss. So the gradient of the uniformity loss on $u_1^o$ can be written as
> $$\frac{\partial L_{\rm{uniform}}}{\partial u_1^o}=\frac{1}{2\tau n}\left(\sum_{v\in{\mathcal{V}/u_1^o}}s_vv\right)+\frac{1}{2\tau n}\left(\sum_{v\in\mathcal{V}/u_1^o}t_vv\right)$$
> It is noteworthy that by writing $\mathcal{V}=\{h_1,h_2,\dots,h_N\}$, Eq(4) shares the same form as the above equation. By adopting the stop-gradient method just as [4] takes, we remove the second term in Eq(4), which is just the $\mathbf{A}\mathbf{D}^{-1}$ term in Eq(6).
>
>
> Empirically, we draw the singular value distribution of embeddings for vanilla BERT and +ContraNorm with only $\mathbf{A}\mathbf{D}^{-1}$ term or $\mathbf{D}^{-1}\mathbf{A}$ term. The figure is attached to Appendix A in the revised paper and can be also viewed through [https://ibb.co/BGQjCSw](https://ibb.co/BGQjCSw). As shown in the figure, compared with vanilla BERT with a long-tail distribution (dimensional collapse), adding ContraNorm with $\mathbf{A}\mathbf{D}^{-1}$ and $\mathbf{D}^{-1}\mathbf{A}$ both reduce the number of insignificant (nearly zero) singular values and make a more balanced distribution. The similar singular value distributions mean they play a similar role in alleviating dimensional collapse.
>
> **The explanation of the stop-gradient operation can be found in Appendix A of our revised paper.**
>
> [3] A Simple Framework for Contrastive Learning of Visual Representations, ICML 2020
>
> [4] Exploring the Equivalence of Siamese Self-Supervised Learning via A Unified Gradient Framework, CVPR 2022
>
> ---
>
> Thanks for your comments and hope our answers could address your concerns. We have also revised our paper accordingly. Please let us know if you have additional questions.

---

> ### Author Response · Authors · 2022-11-15
> **Response to reviewer HpT5 (3/4)**
>
> **Q4.** I do not understand how the layer normalization can be replaced by the factor $1+s$ in Eq.9.
>
> **A4.** We have re-organized this part (Section 3.3) to state it more clear in our revised version. We denote $\mathbf{H}_b$ for the representation matrix before the update and $\mathbf{H}_t$ for the representation matrix after the update. In the new version, in order to solve the problem that the scale of $\mathbf{H}$ may become smaller after the update, we propose two methods:
> - **a)** The first method is to add a regularization term $-\frac{1}{2}\sum_i\|h_i\|_2^2$ to the uniformity loss. When the regularization term becomes smaller, the norm of $h_i$ becomes larger. Therefore, adding this term can help prevent the norm of representations $h_i$ from becoming smaller. In this way, the update form becomes
> \begin{equation}
>     \mathbf{H}_t = (1 + s) \mathbf{H}_b - s/\tau \times \text{softmax}(\mathbf{H}_b\mathbf{H}_b^{\top})\mathbf{H}_b.
> \end{equation}
> We then conduct the theoretical analysis on this update form, showing the fact that it will make the representations more uniform.
>
> - **b)** The second method is to leverage layer normalization, which adjusts the representation according to its mean and variance. The update form of the layer normalization on representation $h_i$ is
> $$\text{LayerNorm}(h_i)=\gamma\cdot\frac{h_i-\text{mean}(h_i)}{\sqrt{\text{Var}(h_i)+\varepsilon}}+\beta,$$
> where $\gamma$ and $\beta$ are learnable parameters and $\varepsilon=10^{-5}$ prevents the denominator from becoming zero. The learnable parameters $\gamma$ and $\beta$ can rescale the representation $h_i$ to help ease the problem. We append the layer normalization as:
> \begin{equation}
>     \mathbf{H}_t = \mbox{LayerNorm}\left(\mathbf{H}_b - s/\tau \times \text{softmax}(\mathbf{H}_b\mathbf{H}_b^{\top})\mathbf{H}_b\right),
> \end{equation}
> where applying the layer normalization to a representation matrix $\mathbf{H}$ means applying the layer normalization to all its components $h_1,\dots,h_n$.
>
> We empirically compare the two proposed methods and find their performance comparable, while the second one performs slightly better, which is shown in Table 4 in the paper. Therefore, we adopt the second update form and name it Contrastive Normalization (**ContraNorm**).
>
> ---
>
> **Q5.** The $\mathbf{H}$ in Eq.6-12 is not clear.
>
> **A5.** These equations all describe how the representaion matrix $\mathbf{H}$ is updated. We revise these equations by using $\mathbf{H}_b$ for the representation matrix before the update and $\mathbf{H}_t$ for the representation matrix after the update.
>
> ---
>
> **Q6.** How are the methods actually implemented in the Transformers. How the stop-gradient is used? How the layer normalization is configured in terms of the bias and scale factor? How is the former layer normalization dealt with when you apply the contranorm before it?
>
> **A6.** A general update of $\mathbf{H}^{(l)}$ in $l$-th block of  Transformers shows as
>
> \begin{align}
> \mathbf{H}^{(l)} &= \text{MultiHeadAttention}(\text{LayerNorm}(\mathbf{H}^{(l-1)})), \tag{R1}\\\\
> \mathbf{H}^{(l)} &= \text{LayerNorm}(\mathbf{H}^{(l)} + \mathbf{H}^{(l-1)}), \tag{R2}\\\\
> \mathbf{H}^{(l+1)} &= \text{MLP}(\mathbf{H}^{(l)}) + \mathbf{H}^{(l)}. \tag{R3}
> \end{align}
>
> In practice, we plug ContraNorm between the second LayerNorm and MLP module, namely, between Eq.(R2) and Eq.(R3). Note that our ContraNorm also has a layer normalization component, so naturally we absorb the original LayerNorm into ContraNorm. Formally, the adapted Transformers is written as
>
> \begin{align}
> \mathbf{H}^{(l)} &= \text{MultiHeadAttention}(\text{LayerNorm}(\mathbf{H}^{(l-1)})),\\\\
> \mathbf{H}^{(l)} &= \text{ContraNorm}(\mathbf{H}^{(l)} + \mathbf{H}^{(l-1)}), \\\\
> \mathbf{H}^{(l+1)} &= \text{MLP}(\mathbf{H}^{(l)}) + \mathbf{H}^{(l)}.
> \end{align}
>
> We keep other components of Transformers unchanged, including the first LayerNorm. The configuration of the second LayerNorm, which is absorbed into our ContraNorm, is also the same as original setting.
>
> (more on A6 below)

---

> ### Author Response · Authors · 2022-11-15
> **Response to Reviewer HpT5 (2/4)**
>
> **Q2.** An analysis of computational cost of the proposed ContraNorm.
>
> **A2.** The formulation of ContraNorm is written as
>
> $$\text{ContraNorm}(\mathbf{H}) = \text{LayerNorm}(\mathbf{H} - s \times \text{softmax}(\mathbf{H}\mathbf{H}^{\top})\mathbf{H}).$$
>
> The most time-consuming operation is the matrix multiplication between $\mathbf{H}$ and $\mathbf{H}^{\top}$. Given $\mathbf{H} \in {\mathbb{R}^{n \times d}}$, where $n$ and $d$ denote the number of samples in a batch and feature size respectively, the time complexity of ContraNorm is $O(n^2d)$, which is the same order as the self-attention operation in Transformer. Empirically, we report the training time of BERT with or without ContraNorm on GLUE tasks in the following table. All experiments are conducted on a single NVIDIA GeForce RTX 3090. On average, we raise the performance of BERT on GLUE tasks from 82.59% to 83.54% (see Table 1 in the paper) with less than 4 minutes overhead. We think the time overhead is acceptable considering the benefits it brings.
>
> |           |  RTE  |  COLA |  MRPC |  STSB |  SST2  |  QNLI  |  MNLI-m / MNLI-mm |    QQP   |  Avg   |
> |:---------:|:-----:|:-----:|:-----:|:-----:|:------:|:------:|:-----------------:|:--------:|:------:|
> |    BERT   | 1m14s | 1m50s | 1m14s | 1m50s | 14m11s | 40m02s |      2h16m26s     | 1h47m38s |  38m03s |
> |+ContraNorm| 1m20s | 2m05s | 1m20s | 2m02s | 16m23s | 43m14s |      2h29m01s     | 1h59m10s | 41m49s |
>
>
> ---
>
> **Q3.** In the ImageNet1K experiment, the performance for $L=24$ is still lower than that of $L=16$, which challenges the mechanism of feature collapsing discussed in this paper.
>
> **A3.** Oversmoothing is a phenomenon where the performance of Transformers or GNNs degenerates greatly when the layer goes deeper. We find in deep layers, the representations undesirably gather in a low-dimensional manifold, in other words, dimensional collapse happens. Therefore, we design ContraNorm to draw the representations far away from each other with the aim of a uniform distribution. **Thus, in the same deep layer, compared between adding ContraNorm before and after (vertical comparison), we effectively alleviate dimensional collapse and prevent performance from deteriorating**, e.g., on ImageNet1K, after adding ContraNorm, the performance of 16-layer ViT and 24-layer ViT are 79.04% and 78.67%, respectively, both higher than vanilla ViT with the same layer.
>
> As you pointed out, when **comparing horizontally** on ImageNet1K, after adding ContraNorm, the performance of 24-layer ViT 78.67% is lower than 79.04% for 16-layer ViT. **The reason is that deeper layers present more severe dimensional collapse, but we keep the intensity of ContraNorm (namely, scale factor $s$) as a constant across all layers, which may be relatively small for deep layers**. Therefore, in the above example, the effect of ContraNorm on alleviating dimensional collapse after 16 layers is not as strong as in the preceding layers. To verify this, for 24-layer ViT on ImageNet1K, we maintain the scale factor of ContraNorm as 0.3 in 1-16 layers, but raise it to 0.4 in 17-24 layers. In such setting, we get a test accuracy of 79.14%, which is higher than 79.04% for 16-layer ViT with ContraNorm.
>
>
>
> |                                             | #L=12 | #L=16 | #L=24 |
> | ------------------------------------------- | :---: | :---: | :---: |
> |                  DeiT-small                 | 77.32 | 78.25 | 77.69 |
> |        + ContraNorm (constant $s$)          | 77.80 | **79.04** | **78.67** |
> | + ContraNorm (higher $s$ for deeper layers) |   -   |   -   | **79.14** |
>
>
> ---

---

> ### Author Response · Authors · 2022-11-15
> **Response to Reviewer HpT5 (1/4)**
>
> We thank Reviewer HpT5 for careful reading and detailed reviews. We address your concerns in the following and revise our paper accordingly.
>
> ---
>
> **Q1.** Why is it not possible to apply uniformity loss straightforwardly? The authors should provide a comparison between the proposed method and uniformity loss.
>
> **A1.** As you suggested, we conduct an additional comparative experiment on BERT model. Specifically, we add the uniformity loss ($loss_{\text{uni}}$) straightforwardly to the classification loss (MSELoss, CrossEntropyLoss, or BCELoss depending on the task type, denoted by $loss_{\rm{cls}}$). Formally, the final loss is written as
>
> $$loss_{\rm{total}} = loss_{\rm{cls}} + loss_{\rm{uni}},$$
>
> where $loss_{\rm{uni}} = \sum_{i=1}^{N} \log \sum_{j=1}^N \exp(f(\mathbf{x}_i)^Tf(\mathbf{x}_j) / \tau)$ and $N$ is the number of samples in the batch. We tune $\tau$ in the range of $[0.5, 0.8, 1.0, 1.2, 1.5]$ and choose the best one in terms of average performance. Other hyperparameters are kept the same as settings of ContraNorm. The results are shown in the following table.
>
>
> |    | RTE | COLA | MRPC| STSB | SST2 |QNLI|MNLI-m|MNLI-mm|QQP|Avg|
> |:--:|:--:|:--:|:--:|:--:|:--:|:--:|:--:|:--:|:--:|:--:|
> |BERT| 68.59 | 55.28 | 88.96 | 88.61 | 92.89 | 91.51 | 84.65 | 84.61 | 88.24 | 82.59 |
> |+Uniformity Loss| 68.59 | 58.08 | 89.46 | **88.69** | 93.00 | 91.60 | 84.45 | 84.43 | 88.14 | 82.94 |
> |+ContraNorm| **70.76** | **58.83** | **89.49** | 88.66 | **93.12** | **91.78** | **84.87** | **84.66** | **88.3** | **83.39** |
>
>
> We can see that `+ContraNorm` gets the best score in 8 / 9 tasks, while `+Uniformity loss` reaches the best only in 1 / 9 tasks. ContraNorm also has the highest average score among all tasks. The reason is that updating the total loss is a combined process for objectives of correct classification and uniform distribution. Thus, a lower $loss_{\rm{total}}$ may be only caused by a lower classification loss while uniformity loss is kept the same, which cannot ensure a more uniform distribution of representations. In contrast, ContraNorm acts directly on representations in each layer and enforces the uniform property.
>
> In fact, there are many methods in GNNs such as [1] and [2], which design the propagation mechanism under the guidance of the corresponding objective. The well-designed propagation mechanism is shown to be the most fundamental part of GNNs. Instead of directly using the loss function, these methods transfer the loss function into a specific propagation method and achieve superior performance, which indicates that changing the network may be more effective than directly adding the objective to the loss function.
>
> [1] Graph Neural Networks Inspired by Classical Iterative Algorithms, ICML 2021
>
> [2] Interpreting and Unifying Graph Neural Networks with An Optimization Framework, ACM 2021
>
> We have added this discussion in the revision (**Appendix D**).
>
> ---

---

### Author Response · Authors · 2022-11-16
**A Summary of Paper Updates**

We thank all reviewers for their constructive comments. We have updated the paper accordingly with the following major changes:

- **Section 1.** Update Figure.1 for illustrating how our proposed ContraNorm solves dimensional collapse.

- **Section 2.** Add related work about *whitening methods for dimensional collapse*.

- **Section 3.1.** Add a comparison between the singular value distributions of embeddings on BERT w/ and w/o ContraNorm.

- **Section 3.3.** Update the denotation $\mathbf{H}$ to $\mathbf{H}_b$ (embeddings before ContraNorm) and $\mathbf{H}_t$ (embeddings after ContraNorm) in equations. Reformulate statement on layer normalization and scale factor $(1+s)$.

- **Section A.** Add illustration on the stop-gradient in the derivation of ContraNorm.

- **Section D.** Add a detailed comparison between our ContraNorm and straightforwardly applying the uniformity loss.

---

### Decision · Program_Chairs · 2023-01-20

**Decision:**

Accept: poster

**Justification For Why Not Higher Score:**

Three reviewers suggest marginally above the acceptance threshold, while one reviewer likes the idea to convert the gradient to the form of normalization and suggests acceptance. So it is a solid acceptance for poster.

**Justification For Why Not Lower Score:**

All reviewers unanimously suggest to accept the paper.

**Metareview: Summary, Strengths And Weaknesses:**

This manuscript addresses the overs-moothing problem of graph neural network (GNN) and Transformers, and proposes to avoid the dimensional collapses in the representation, borrowing the idea from self-supervised learning. It proposes a normalization layer ContraNorm, which aims to learn a more uniform distribution in the embedding space that can alleviate dimensional collapse, motivated on the uniform loss from self-supervised learning. The proposed normalization layer can be inserted into GNNs and Transformers, and improves these baselines based on the experiments on three different scenarios: ViT for image classification, BERT for natural language understanding, and GNNs for node classifications. Reviewers unanimously suggest to accept the paper, so is the decision.

**Note From Pc:**

if the above contains the word "oral" or "spotlight" please see: "oral" presentation means -> notable-top-5% and "spotlight" means -> notable-top-25%. As stated in our emails, we are disassociating presentation type from AC recommendations